# Highly structured populations of deep-sea copepods associated with hydrothermal vents across the Southwest Pacific, despite contrasting life history traits

Coral Diaz-Recio Lorenzo[1,2,3]*, Tasnim Patel[4], Eve-Julie Arsenault-Pernet[5], Camille Poitrimol[3,5], Didier Jollivet[3], Pedro Martinez Arbizu[6], Sabine Gollner[1]

1 NIOZ Royal Netherlands Institute for Sea Research, 't Horntje (Texel), The Netherlands, 2 Utrecht University, Utrecht, The Netherlands, 3 Adaptation et Diversité en Milieu Marin (AD2M), Station Biologique de Roscoff, Sorbonne Université, CNRS, Roscoff, France, 4 Royal Belgian Institute of Natural Sciences, Brussels, Belgium, 5 Biologie et Ecologie des Ecosystèmes marins Profonds (UMR BEEP UBO-CNRS-IFREMER), IFREMER Centre de Bretagne, Plouzané, France, 6 Senckenberg am Meer, German Center for Marine Biodiversity Research, Wilhelmshaven, Germany

* coral.diazrecio@nioz.nl

**Data Availability Statement:** All accession numbers are found in supplementary information

## Abstract

Hydrothermal vents are extreme environments, where abundant communities of copepods with contrasting life history traits co-exist along hydrothermal gradients. Here, we discuss how these traits may contribute to the observed differences in molecular diversity and population genetic structure. Samples were collected from vent locations across the globe including active ridges and back-arc basins and compared to existing deep-sea hydrothermal vent and shallow water data, covering a total of 22 vents and 3 non-vent sites. A total of 806 sequences of mtDNA from the *Cox1* gene were used to reconstruct the phylogeny, haplotypic relationship and demography within vent endemic copepods (Dirivultidae, *Stygiopontius* spp.) and non-vent-endemic copepods (Ameiridae, Miraciidae and Laophontidae). A species complex within *Stygiopontius lauensis* was studied across five pacific back-arc basins at eight hydrothermal vent fields, with cryptic species being restricted to the basins they were sampled from. Copepod populations from the Lau, North Fiji and Woodlark basins are undergoing demographic expansion, possibly linked to an increase in hydrothermal activity in the last 10 kya. Highly structured populations of *Amphiascus* aff. *varians* 2 were also observed from the Lau to the Woodlark basins with populations also undergoing expansion. Less abundant harpacticoids exhibit little to no population structure and stable populations. This study suggests that similarities in genetic structure and demography may arise in vent-associated copepods despite having different life history traits. As structured meta-populations may be at risk of local extinction should major anthropogenic impacts, such as deep-sea mining, occur, we highlight the importance of incorporating a trait-based approach to investigate patterns of genetic connectivity and demography, particularly regarding area-based management tools and environmental management plans.

S1 alongside the metadata for each sample including geographic sampling locations and the equipment used to collect the samples. Data pertaining to shallow water harpacticoid copepods is available within the manuscript (Fig 3) and also in S1 and GenBank, with accession numbers: MH670513, MH670516, MH708087, MH242650, MH642651, MH670524, MH670508, MH670525, MH670521, MN507530, MN242965, MN542380, MN542379. Data pertaining to deep-sea harpacticoids generated for this study are available within the manuscript (Fig 3) and also in S1 and GenBank with accession numbers: OQ693533-OQ693549 and OQ693550-OQ693585 Data pertaining to dirivultid copepods can be found in S1 and GenBank with accession numbers: GQ925954-GQ926174 KX714850-KX714908 MT934831-MT934839 OQ692990-OQ693580.

**Funding:** This work was supported by the Alexander von Humboldt Foundation (to Sabine Gollner), and by UU-NIOZ under the project "Protecting deep-seabed hydrothermal vents via area-based management tools" (to Sabine Gollner). Ship time for the TN235 expedition was funded by the US National Science Foundation (NSF) grants OCE- 0732333 to Charles R. Fisher and OCE-0732439 to George W. Luther III. Ship time during the CHUBACARC expedition was supported by the French Oceanographic Fleet programme (CHUBACARC cruise https://doi.org/10.17600/18001111 to Didier Jollivet and Stephane Hourdez), INEE (CNRS) and the Agence Nationale de la Recherche ANR "CERBERUS" (contract number ANR-17CE02-0003 to Stephane Hourdez). The funders had no role in study design, data collection and analysis, decision to publish, or preparation of the manuscript.

**Competing interests:** The authors have declared that no competing interests exist.

## Introduction

Deep-sea hydrothermal vents are ephemeral and highly productive ecosystems inhabited by abundant fauna that often depend on chemosynthetic primary production, fueled by the energy provided by vent fluid emission [1,2]. The active and diffuse vents are environments governed by tectonic seafloor spreading which gives rise to gradients in temperature, concentrations of hydrogen sulfide, methane, oxygen, and heavy metals [3–5]. Mega-, macro- and meiofaunal communities are structured along these gradients, inhabiting niches based on their thermo-tolerance, adaptations to low oxygen levels and high concentrations of hydrogen sulfide, as well as their ability to tolerate elevated levels of toxic metals [6–8]. This patchiness of the vent environment gives rise to metacommunities and metapopulations connected by, but not exclusively, pelagic, and planktonic larval dispersal [9,10]. Most studies investigating faunal dynamics at vents have until now focused on megafauna assemblages such as *Riftia* and *Paralvinella* tubeworms, *Bathymodiolus* mussels, *Ifremeria* and *Alviniconcha* snails, *Shinkaia* squat lobsters, *Rimicaris* shrimp and more, and have focused on zonation [11], nutritional resources [12] diversity and connectivity [13], traits [14], ecotoxicology [15] larval ecology [16] and adaptations to vent conditions [17].

Despite being estimated to account for over 50% of the metazoan biodiversity at hydrothermal vents [18], studies on meiofauna are still rare. Where hard substrate is available (i.e., basalt rock or megafaunal assemblages that create landscapes of exoskeletons and shells), the Copepoda are the most abundant taxon. Of these, the Dirivultidae (Siphonostomatoida), a family of deep-sea, vent-endemic copepods, are by far the most abundant meiofauna, existing in the tens to hundreds of thousands, and contributing to >90% of the community abundance with one or two dominant species [19–22]. This specialised family has therefore adapted to thrive in typical vent conditions [19,23–25] (Table 1), occupying niches in the more extreme end of environmental gradients. The low end of the spectrum, where conditions are more similar to the ambient environment (i.e., higher oxygen levels and lower temperatures), is occupied by copepods that also exist in non-vent environments, peripheral vent environments and shallow waters. This low end of the spectrum is mostly comprised of the order Harpacticoida with families such as the Miraciidae, Ameiridae and the Laophontidae, which are commonly observed in diffuse vent environments [20,21,26,27]. The diversity of these copepods has been to date large unexplored, with only a few studies investigating their community structure, habitat preferences, and molecular diversity. Furthermore, until now, the latter has solely been used to explore the genetic differences of Dirivultidae [19,21,25,28,29]. These studies have incorporated taxonomic, molecular, metabolic, and experimental approaches to investigate the diversity of vent copepods in the Mid Atlantic Ridge and the Central Indian Ridge [19,20,25,28,29] the East Pacific Rise [19,28], Northwest Pacific [22,28], and Southwest Pacific [21].

Traits are increasingly being used to quantify global biodiversity patterns, with trait databases growing in size and number, across diverse taxa [14]. Trait-based approaches are now common frameworks for understanding ecosystem processes in the marine environment, from microbial to megafaunal communities [43–45]. A distinguishing and pivotal feature between the groups of copepods associated with deep-sea hydrothermal vents, namely Dirivultidae, Ameiridae, Miraciidae and Laophontidae, is the differences in their life history traits. We define and outline traits as per the sFDvent trait database [14]. These traits include relative adult mobility, depth range (m), chemosynthesis-obligate, estimated maximum body size (mm), zonation from a vent, substratum, habitat complexity, gregariousness (frequency with which it is found in groups/clusters), trophic mode, nutritional mode, foundation species, and abundance (Table 1). We also include the dispersal type, lecithotrophic (dependent on an egg yolk during larval dispersal) or planktotrophic (larvae do not have a yolk but can feed during

**Table 1. Traits as described in sFDvent.**

| Trait Category | Trait | Dirivultidae (*Stygiopontius*) | Miraciidae | Ameiridae | Laophontidae | Reference |
|---|---|---|---|---|---|---|
| Mobility | Relative adult mobility (speed in mm sec$^{-1}$) | >30 | 0.42–10 | 0.42–10 | 0.42–10 | [25,30] |
| Community structure | Abundance/10 cm$^2$ | ≤ 32 | <10 | <10 | <10 | [21,31,32] |
| | Relative abundance (%) | >90 | <10 | <10 | <10 | [21,31,32] |
| Geographic distribution | Depth range (m) | 700–4000 | 0–4000 | 0–4000 | 0–4000 | [18,21] |
| Generalist/ specialist | Chemosynthetic obligate | Vent-specialist | Generalist | Generalist | Generalist | [19,23–25,28,33,34] |
| Life history | Larval dispersal method | Lecithotrophic | Planktotrophic | Planktotrophic | Planktotrophic | [19,25,35] |
| | Number of eggs / 1 female specimen | 4 | 7–10 | 14 | 6 | [36–39] |
| | Estimated **maximum** body size (mm) | 2 | ≤1 | ≤1 | ≤1 | [21,25] Visual observation |
| Habitat use | Zonation from a vent | Low | Medium/Low | Medium/Low | Medium/Low | [21,40,41] |
| | Substratum | Hard | Soft and Hard | Soft and hard | Soft and hard | [20,24] |
| | Gregariousness | Always | Sometimes | Sometimes | Sometimes | [19–21] |
| Trophic structure | Nutritional Sources | Chemoautotrophic bacteria | Organic detritus, organic bacterial mats | Organic detritus, organic bacterial mats | Organic detritus, organic bacterial mats | [34,40,42] |
| Species associations at vents | Foundation species | *Alviniconcha snails, Ifremeria snails, Bathymodiolus mussels, Paralvinella worms, Rimicaris shrimp, Shinkaia squat lobsters* | *Bathymodiolus* | *Bathymodiolus* | *Bathymodiolus* | [19,21,25] |

A global trait database for deep-sea hydrothermal vent fauna, listing the trait category, the trait within the category, the family for which the trait is described (Dirivultidae, Miraciidae, Ameiridae and Laophontidae), and the references for each family.

dispersal) as well as number of eggs, to shed light on differences in fecundity. We note that the mode of dispersal is often unknown for the nauplii of copepod species. However, whilst many nauplii and copepodite I stages of Dirivultidae have been observed in waters above vents [24], nauplii from the harpacticoid families in this study, are typically benthic-dwelling and do not migrate vertically to disperse in the plankton, but rather are free-living, and benthic dwelling throughout their life history, and may be transported occasionally by bottom or pelagic currents [46]. Life history traits at vents have been largely explored for charismatic megafauna and macrofauna [14,25,47–52] but only a few studies have explored such traits for meiofauna at vents [18,29].

Dirivultid copepods are highly mobile in adult stages, developing specialised swimming and crawling legs during settlement at a vent [24,53]. They have a depth range of 700–4000 m restricting them to deep water hydrothermal vent environments and although they (to our current knowledge) do not have bacterial endosymbionts, are considered vent (chemosynthetic) obligates as they are found in extremely high abundances in areas with typical vent conditions relative to ambient conditions [19]. Dirivultids have a large estimated maximum body size (~1mm, and up to 2 mm) and carry only 4 large yolk-rich eggs per female, from which lecithotrophic nauplii hatch. Despite this low number of eggs however, they could exist in the tens to hundreds of thousands as suggested by species abundance data from copepods associated with megafauna assemblages at the ABE vent field [19,21]. Abundance data of nauplii and

copepodites from plume samples in the East Pacific Rise have revealed higher densities of Dirivultidae (50 individuals / 1000 L, totaling 8657 ind. 94% of which were Dirivultidae copepodites stage 1) relative to harpacticoid copepods (total = 73 ind.) [18]. Dirivultids are found only where the substratum is hard, living on basalt rock and on/amongst assemblages of foundation fauna such as *Alviniconcha* and *Ifremeria* snails and *Bathymodiolin* mussels, in diffuse vent areas as well as on vent chimneys close to the vent source. In some cases, they have been found in high abundance inside the mouths and guts of the vent polychaete worm *Paralvinella sulfincola* [42], which despite not having a predatory feeding morphology, may ingest them alongside other particulate matter when hydrothermal flow is strong. They have also been found in the gills of Alvinocarid shrimp [18], where they may be feeding on symbiotic bacteria. Studies suggest that this family is bacterivore, feeding on chemoautotrophic bacterial mats and films to differing degrees of discrimination [21,34,40–42,51], further suggestive of the fact that these organisms are chemosynthetic obligates. In contrast, harpacticoid members of the Miraciidae, Ameiridae and Laophontidae families exist at vent environments in much lower abundance, found predominantly in ambient benthic sediments and shallow water environments [18,20,21,43,54,55]. Their depth range can be thousands of meters [47], they are habitat generalists, and exist across a broad environmental gradient [24,51]. Most harpacticoids are abundant at the medium-low end of the hydrothermal gradient, where oxygen levels are more tolerable to a wider range of species [21,24]. Harpacticoids are benthic, but have a planktotrophic, free-living larval dispersal, and generally produce more eggs than the Dirivultidae (Table 1) [36–38,56].

Copepod communities are poorly understood despite forming the most abundant and diverse group of animals wherever hard substrate is available at hydrothermal vents, and inhabiting resource-rich ecosystems that are explored for deep-sea mining of Seafloor Massive Sulfides (SMS). In particular, diversity estimates, and data on connectivity and demography of populations are still largely lacking. This study reports the molecular diversity and demography of a new species complex of *Stygiopontius lauensis* in the Southwest Pacific region spanning from the Lau to the Manus back-arc basins (BABs) (Fig 1B) and its relationship to the rest of the *Stygiopontius* genus from BABs and Mid-Ocean Ridges (MORs), based on the mitochondrial Cytochrome c. oxidase subunit I gene (*Cox1*). We also report the molecular diversity and demography for species within the following families, Miraciidae (*Amphiascus* aff. *varians* 1, and 2, and *Sarsamphiascus* sp. 1), Ameiridae (*Ameira* sp. 4 and 5), and Laophontidae (*Bathylaophonte pacifica*), and discuss the integral role of life traits for the observed diversity and demography.

## Materials and methods

### Sampling area

The main study area covered ten vent fields, Tu'i Malila, ABE, Tahi Moana, Tow Cam, Kilo Moana, Mangatolo, Fatu Kapa, Phoenix, La Scala, and Pacmanus/Solwara 8 from five basins in the Southwest Pacific Ocean (Lau, Futuna Volcanic Arc, North Fiji, Woodlark, and Manus) (Fig 1A and 1B). Data was compared to existing samples collected from Eastwall, Tica, Bio-9, V-Vent, Marker-28, and P-Vent, from the East Pacific Rise (EPR), Rebecca's Roost from the Guaymas Basin, TAG and Snake Pit from the Mid Atlantic Ridge (MAR), Sakai from the Okinawa Trough, and Bayonnaise and Myojin-sho from the Izu-Bonin-Mariana Arc(IBM-arc), covering a total of 22 vent fields. Shallow water data from the North Sea, a littoral zone around Hawaii, and the San Juan Islands were also included (Table 2).

### Sample collection

Copepods were collected during several cruises from 2009 to 2019 to the Lau Basin, Futuna Volcanic Arc, North Fiji, Woodlark and Manus basins (for expedition details please refer to

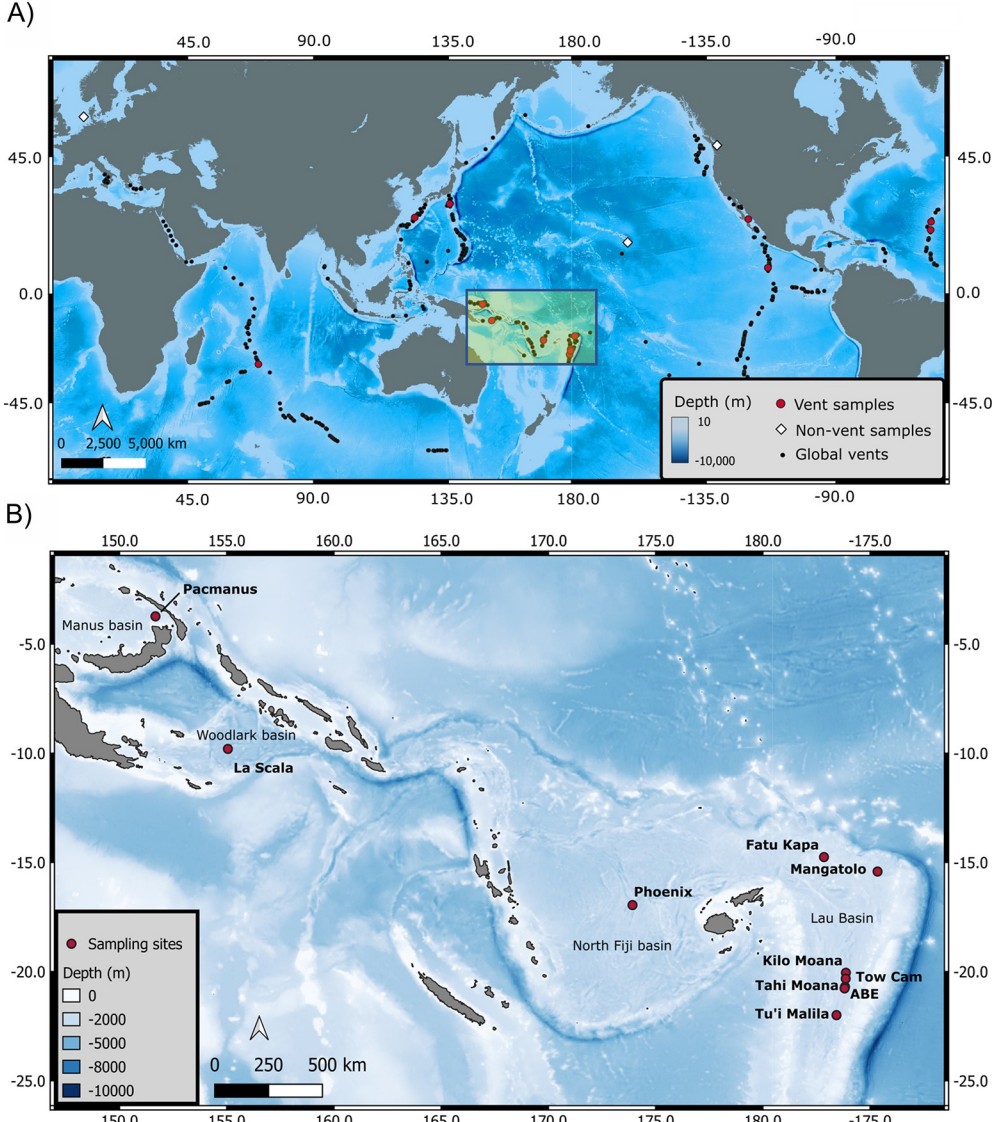

**Fig 1. Sampling area.** a) Map of global study sites where, white diamonds = non-vent samples, red circles = vent samples, black circles = all globally distributed vents. B) Detailed distribution of sampling sites (10 vents) within the Southwest pacific back-arc basins, including the Manus, Woodlark, North Fiji and Lau Basins and the Futuna Volcanic Arc.

[18,19,57,58]). Copepod specimens were collected using human operated vehicles *Alvin* on the EPR and Guaymas Basin (GB), the *Nautile* on the Mid-Atlantic Ridge (MAR), the remotely operated vehicle (ROV) *Kiel6000* along the Central Indian Ridge (CIR) and the ROVs *Jason* and *Victor6000* in cruises in 2009 and 2019 to the Southwest Pacific, respectively. Megafaunal assemblages were collected using mussel pots and slurp guns as well as grabs using the robotic arm. On board, megafaunal samples were washed and meiofauna were retained on 32 μm sieves and fixed in 99% EtOH during the 2009 and 2016 expeditions, and in 2019, meiofauna were retained on 250 μm mesh, fixed in 96% EtOH. All samples were placed in cold storage immediately after collection (4°C). Samples from 2009 were sorted and processed at the German Center for Marine Biodiversity Research (DZMB), while those from 2016 were brought

**Table 2. Species and sampling site information.**

| Sampling site | North Sea | Hawaii (littoral) | San Juan Islands | Mangatolo | Kilo Moana | Tow Cam | Tahi Moana | ABE | Tu'i Malila | Fatu Kapa | Phoenix | La Scala | Pacmanus | Eastwall | Tica | Bio_9 | V-Vent | Marker_28 | P-Vent | Rebecca's Roost | TAG | Snake_Pit | Kairei (INDEX13-31ROV) | Kairei (INDEX13-35ROV) | Sakai | Myojin-sho | Bayonnaise |
|---|---|---|---|---|---|---|---|---|---|---|---|---|---|---|---|---|---|---|---|---|---|---|---|---|---|---|---|
| Region | North Sea | Hawaii | NW Pacific | Lau Basin | Lau Basin | Lau Basin | Lau Basin | Lau Basin | Lau Basin | Futuna Arc | North Fiji | Woodlark | Manus | EPR | EPR | EPR | EPR | EPR | EPR | Guaymas Basin | MAR | MAR | CIR | CIR | Okinawa Trough | Izu-Bonin Arc | Izu-Bonin Back-arc |
| Latitude | 55.04 | 21.44 | 48.54 | -15.42 | -20.05 | -20.32 | -20.69 | -20.77 | -21.97 | -14.75 | -16.95 | -9.8 | -3.73 | 9.85 | 9.83 | 9.83 | 9.78 | 9.83 | 9.83 | 27 | 26.12 | 23.37 | -25.19 | -25.19 | 27.55 | 32.1 | 31.97 |
| Longitude | 6.33 | 158 | -123 | -174.66 | -176.13 | -176.14 | -176.18 | -177.18 | -176.57 | -177.15 | 173.91 | 155.1 | 151.67 | -104.3 | -104.28 | -104.28 | -104.28 | -104.28 | -104.28 | -111.4 | -44.82 | -44.95 | 70.02 | 70.02 | 126 | 139.87 | 139.73 |
| Depth (m) | 10–50 | 12–18 | 0 | 2039 | 2620 | 2716 | 2256 | 2153 | 1886 | 1562 | 1961 | 3388 | 1737 | 2500 | 2509 | 2508 | 2508 | 2507 | 2509 | 2000 | 3620 | 3450 | 2527 | 2379 | 980 | 802 | 770 |
| Harpacticoida-Ameiridae | | | | | | | | | | | | | | | | | | | | | | | | | | | |
| Ameira nov. sp. Aff. parvula 1 | 2 | | | | | | | | | | | | | | | | | | | | | | | | | | |
| Ameira nov. sp. Aff. parvula 2 | 2 | | | | | | | | | | | | | | | | | | | | | | | | | | |
| Ameira n. sp. 1 | 4 | | | | | | | | | | | | | | | | | | | | | | | | | | |
| Ameira sp. 4 | | | | | | | | | | | | | | 11 | | | | | | | | | | | | | |
| Ameira sp. 5 | | | | | | | | 6 | | | | | | | | | | | | | | | | | | | |
| Harpacticoida-Miraciidae | | | | | | | | | | | | | | | | | | | | | | | | | | | |
| Amphiascus aff. variants 1 | | | | | | | | | | | | | | 3 | 3 | | | | | | | | | | | | |
| Amphiascus aff. variants 2 | | | | | 2 | 2 | | 13 | | | | 6 | | | | | | | | | | | | | | | |
| Sarsamphiascus sp. 1 | | | | | | | | | 4 | | | | | | | | | | | | | | | | | | |
| Sarsamphiascus hawaiiensis | | 1 | | | | | | | | | | | | | | | | | | | | | | | | | |
| Sarsamphiascus urdoiss | | | 1 | | | | | | | | | | | | | | | | | | | | | | | | |
| Sarsamphiascus kawamurai | | 1 | | | | | | | | | | | | | | | | | | | | | | | | | |
| Harpacticoida-Laophontidae | | | | | | | | | | | | | | | | | | | | | | | | | | | |
| Bathylaophonte pacifica | | | | | | | | | | | | | | 2 | 1 | | | | | | | | | | | | |
| Siphonostomatoida-Dirivultidae | | | | | | | | | | | | | | | | | | | | | | | | | | | |
| Stygiopontius brevispina | | | | | 8 | | | 1 | 26 | | | | | | | | | | | | | | | | | | |
| Stygiopontius lauensis | | | | 77 | 7 | 44 | 32 | 55 | 78 | 77 | | | | | | | | | | | | | | | | | |
| Stygiopontius aff. lauensis 1 | | | | | | | | | | | 62 | | | | | | | | | | | | | | | | |
| Stygiopontius aff. lauensis 2 | | | | | | | | | | | | 85 | | | | | | | | | | | | | | | |
| Stygiopontius aff. lauensis 3 | | | | | | | | | | | | | 35 | | | | | | | | | | | | | | |
| Stygiopontius senckenbergi | | | | | | | | | | | | | | | 7 | 32 | 27 | 16 | | | | | | | | | |
| Stygiopontius hispidulus | | | | | | | | | | | | | | | | | | | | 1 | | | | | | | |
| Stygiopontius sp. 1 | | | | | | | | | | | | | | | | | | | | | | | 17 | | 3 | | |
| Stygiopontius sp. 2 | | | | | | | | | | | | | | | | | | | | | | | 4 | | | | |
| Stygiopontius pectinatus | | | | | | | | | | | | | | | | | | | | | 18 | 15 | | | | | |
| Stygiopontius senokuchiae | | | | | | | | | | | | | | | | | | | | | | | | | | 3 | 3 |

Characteristics of the sampling locations including latitude, longitude. And depth (m) as well as the number of specimens from each species analyzed from each site.

back to the NIOZ laboratory and sorted under a dissecting light microscope. After the 2019 expedition to the Lau Basin, copepods were sorted to family level at IFREMER and further sorted to genus and species level at the NIOZ.

## Taxonomic species identification

Copepods were sorted and identified taxonomically at least to genus level, and to species level where possible. For the Dirivultidae (Siphonostomatoida), only individuals of *Stygiopontius* were used for this study and our dataset comprises eight species: (*Stygiopontius brevispina*, *Stygiopontius lauensis*, *Stygiopontius pectinatus*, *Stygiopontius hispidulus*, *Stygiopontius senckenbergi*, and *Stygiopontius senokuchiae*). Closer examinations were conducted by transferring specimens that had been stored in 99% EtOH to a 3:1 EtOH: glycerin mixture for 20 minutes, followed by a transfer to pure glycerin. Single copepod specimens were then mounted whole on a glass slide and sealed with lacquer. In addition, for *S. lauensis*, at least one female specimen from each vent field was dissected (only females were used in the study due to their large size compared to males and higher DNA yields). Dissection was done with sterile tungsten dissecting needles in a drop of glycerin on a glass slide whilst working with a Leica APO stereomicroscope, and slides were viewed under a Leica light microscope.

## DNA extraction, amplification, and sequencing

The DNA extractions were carried out using Dneasy (Qiagen), Chalex, and the E.Z.N.A. Mollusc DNA extraction Kit (Omega Bio-Tek), following the protocols provided by the manufacturers. The Cytochrome c. oxidase subunit I gene (*Cox1*) was amplified using the Folmer [59] primer set LCO1490 (5'GGTCAACAAATCATAAAGATATTGG'3), and CO2198 (5'TAAACTTCAGGGTGACCAAAAAATCA'3). These primers were ordered with corresponding M13 tails attached: M13F-pUC (-40) (5'GTTTTCCCAGTCACGAC'3) and M13R-pUC (-40) (5'CAGGAAACAGCTATGAC'3), for LCO1490 and CO2198, respectively. Polymerase chain reaction (PCR) cycle conditions were as follows: initial denaturation for 3 minutes at 94˚C followed by 40 cycles of denaturation at 30 seconds for 94˚C, annealing for 45 seconds at 45˚C and extension for 45 seconds for 72˚C and a final extension for 2 minutes at 72˚C. Samples were sequenced with Macrogen Europe.

## Data analysis

A total of 740 sequences from the eight *Stygiopontius* species were used in this study (Table 2). A sum of 552 newly acquired *Cox1* sequences were combined with 188 existing *Cox1* sequences matching the term *Stygiopontius* from NCBI [28,29]. A total of 66 sequences were used for the harpacticoid copepods. Newly acquired sequences for the *Ameira* (17), *Amphiascus* (33), Bathylaophonte (3), were combined with GenBank sequences matching the terms *Ameira* (9) and *Sarsamphiascus* (4). No existing sequences were found for *Bathylaophonte* (S1 File). Sequences from shared genera in shallow water were included only in the phylogenetic construction of the Harpacticoida. All raw sequences were processed (trimmed and assembled into contigs) using Geneious Prime, version 2022.0.2. Ambiguities were resolved by eye from chromatograms, and gaps removed where possible, using forward and reverse sequences. Sequences with a HQ score below 40% were removed. Cleaned contigs from the *S. lauensis* species were batch-blasted against the NCBI nucleotide collection and optimised for highly similar sequences using the Megablast option. Any sequences not pertaining to Dirivultidae were removed, representing contamination, resulting in an amplification success rate of 77%. The remaining sequences had E-values of 0 and matched with *S. lauensis* as the first hit. Sequences were aligned using the MAFFT (version 7.490) plugin in Geneious [60] and cleaned

manually within the Geneious interface. The subsequent alignment was translated into protein sequences using the translation table for invertebrate mitochondria, allowing reading frame identification of each nucleotide sequence separately to check for stop codons following methods from [19]. As no stop codons were found, the alignment was deemed suitable for downstream analysis. Sequences shorter than 500 bp were removed from the alignment.

## Phylogenetic tree construction

Alignments were partitioned into coding frames 1, 2, and 3, to account for codon-usage bias, which if not modelled correctly can lead to erroneous phylogenies [61–65]. Model selection for each partition was carried out in IQ-TREE [66] webserver: http://iqtree.cibiv.univie.ac.at/) for Maximum Likelihood (ML) trees and with BEAST Model Test for Bayesian Posterior Probability (BPP) trees (BEAST 2.6.7). For the harpacticoid alignment, the best evolutionary models selected with Bayesian Information Criterion (BIC) were TIM2+F+I+G4, F81+F+I, and TN+F+G4 for partitions 1, 2 and 3, respectively. For the dirivultid alignment, the models used were TN+F+G4, TPM3+F+G4, and TN+F+G4 for partitions 1,2 and 3, respectively. Duplicate sequences were removed prior to tree construction to avoid inflation of support values and as such the trees were only run on haplotypes. Trees were run with 1000 bootstraps and only the highest likelihood tree is displayed. Bayesian posterior probabilities were calculated using unlinked site models, unlinked clock rates, and linked trees (Fig 1 in S3 File), allowing different evolutionary rates for each partition. Site models were found separately using the Beast Model Test and applied automatically to each partition alongside a strict clock of 1.0. Parameters of the multispecies coalescent were 1.0 for the population mean and linear with constant root for the population function (ploidy was set to mitochondrial). For the priors, the tree construction model was the Yule model, and all standard parameters were retained. A chain length of 200 million, sampling frequency 5000, and burnin of 10% was used. Harpacticoid species were used as the outgroup for the Dirivultidae phylogeny (accession numbers = KX714909, KX714910 and KX714912) and dirivultid species were used as the outgroup for the harpacticoid phylogeny (accession numbers = GQ926008-10), proving to be sufficiently divergent to resolve the trees in both datasets, whilst being sufficiently similar so as to retain topological resolution. All trees were annotated and edited using the interactive Tree of Life (iTOL) (https://itol.embl.de/).

## Species delimitation

Species were delimited using a combination of distance-based and tree-based methods and applied to both the global harpacticoid alignment and the alignment containing sequences from *S. lauensis* from all collections (all other Stygiopontius species have been described and delimited prior to this study). Firstly, the Assemble Species by Automatic Partitioning (ASAP) method [67] was used, which finds the first significant barcoding gap that exists when the divergence between conspecific individuals is smaller than that of individuals of different species from an alignment. Secondly, the Bayesian Poisson Tree Process (bPTP) [68] was used, a process that models branching events in a phylogenetic tree in terms of number of substitutions. This was done using both the ML trees and the BPP trees produced from the alignments in Newick format, and the default parameters from the PTP webserver (MCMC chain = 500,000, thinning = 100, and burnin = 10%). Delimitation results did not differ when using ML or BPP starting trees. Thirdly, Bayes Factor Delimitation (BFD), using nested sampling to calculate the difference between two models, were also used to delimit species from their global alignments. In BFD, the first model hypothesizes that a single species is present and in the second, that multiple species may be present. The Bayes factors were calculated

using the following formula:

$$-mL(1species)-(-mL(2species)) > [SD(1species) + SD(2species)]*2$$

Where mL stands for marginal likelihood and SD stands for standard deviation. If the difference in the two marginal likelihoods is greater than twice the sum of the standard deviation, the model selection favours the multiple species hypothesis. Population analysis with reticulate trees (PopART v.1.7) [69] was used to create minimum spanning haplotype networks. Finally, putative species sequence alignments were subset from the main dataset for subsequent diversity and demographic analyses.

## Analysis of Molecular Variance (AMOVA)

**Diversity.** Pairwise, permutational analyses of molecular variance (AMOVA), was conducted on each putative species, within which specimens were defined by geographic origin using DnaSP v.6 [70]. Analysis of Molecular Variance (AMOVA) was conducted in Arlequin 3.5.2.2 [71], to calculate population specific parameters including haplotype diversity ($Hd$), which describes the probability that two, randomly sampled alleles are different in a population and nucleotide diversity ($\pi$), representing the average number of nucleotide differences per site. All diversity and demographic analysis were conducted on populations with at least three specimens and on populations defined. Net divergence (Da) between populations of the *S. lauensis* were calculated using DnaSP6.

**Demography.** Two neutrality test parameters were also calculated in Arlequin using the infinite site model and 1000 simulations. Firstly, Tajima's $D$ [72], which calculates the difference between the mean number of pairwise differences and the number of segregating sites, and secondly, Fu's $F_s$ [73], which estimates the probability of observing a random sample with numbers of alleles equal to or smaller than the observed value, given the observed diversity and assumption that all the alleles are selectively neutral. These parameters in combination are affected by changes in population size, where negative values of both indicate an excess of low frequency polymorphisms (rare alleles), as expected from a recent population expansion or genetic hitchhiking, causing deviation from mutation-drift equilibrium [74–78]. Sum of Squared Deviations (SSD) and Harpending's raggedness index ($rg$) [79], are parameters that describe the mismatch distribution of pairwise nucleotide differences. The shape of these parameters describes the state of the population i.e., whether it is undergoing expansion or a bottleneck [80]. Non-significant values of $rg$ suggest a good fit of the expansion model to the empirical data, whereas significant values are indicative of constant population size through time, represented as a "ragged" or multi-modal mismatch distribution [81].

To further investigate population growth or bottleneck events within the *Cox1* gene. Extended Bayesian Skyline Plots (EBSP) [82] were constructed in BEAST2 for each cryptic species alignment of *S. lauensis* following the tutorial https://evomics.org/wp-content/uploads/2018/01/ebsp2-tut.pdf. Bayesian Skyline analysis is based on a division of time between the root of a phylogenetic tree and the present time (time to the most recent common ancestor), into segments. Different Effective Population Size ($N_e$) estimates were then calculated for each segment. The endpoints of each segment are tied to branching times (coalescent events) in the tree and the size of each segment is representative of the number of branching events in each segment. Coalescent events are then grouped into segments and $N_e$ and segment size are then jointly calculated [83]. A strict molecular clock was calculated based on [19], who reported a mutation rate of $1 \times 10^{-8}$ events/bp/generation and a generation turnover of 33 generations/year for organisms living at ~20°C. Based on this information, a strict clock of 0.3 mutations/bp/ Ma was used to calibrate the coalescent EBSP for each species. The input parameters for each

run (same for each cryptic species of *S. lauensis*) were as follows: the site model used was input from the previous BEAST Model Test (GTR, Gamma count = 4, shape = 0.82, proportion of invariant sites = 0.66), the clock model was set to 0.3, the priors were set to Coalescent Extended Bayesian Skyline for the tree model and the rest were modified from the tutorial above to achieve convergence (gamma shape = exponential, popSizes.Alltrees = Weibull). All other parameters were set to default. Each run was conducted with 200 million iterations, a pre-burnin of 10%, and stored every 1000 generations.

## Results

### Morphological identification of the *Stygiopontius lauensis* species complex

One morphospecies was identified as *S. lauensis* across the Lau Basin, Futuna Volcanic Arc, North Fiji, Woodlark and Manus basins. For the Ameiridae (Harpacticoida), two morphospecies were identified (*Ameira* sp. 4 and 5) from the East Pacific Rise and the Lau basin, respectively. For the Miraciidae (Harpacticoida), two species were identified, *Amphiascus*. aff. *varians* and Miraciidae sp. 1. Lastly, for the Laophontidae (Harpacticoida), one species was identified, namely *Bathylaophonte pacifica*. For general species identification see [21,24]. Morphological identification of *S. lauensis* female specimens revealed that no morphological differences could be detected amongst specimens from the Lau, Futuna Volcanic Arc, North Fiji, Woodlark and Manus basins. All specimens shared the characters that discriminate the *S. lauensis* species from other *Stygiopontius* species [24,84]. This includes the setation of the 3[rd] exopodal segment of leg 4 (II, I, 4 in *S. lauensis*), the number of coxal setae (on leg 1 and leg 2 in *S. lauensis*), and the additional features of having a smooth leg 1 basis, and spinules on the caudal rami. In addition, all other copepod appendices were similar to the original *S. lauensis* description by [84] (A1, A2, Mnd, MaxI, MaxII, Mxp, leg1–leg5). Furthermore, body size and shape were similar, as well as the ornamentation on the anal segment. The other *Stygiopontius* species had been morphologically identified previously by [28].

### Dirivultidae

**Phylogenetic reconstruction and species delimitation.** The phylogeny of *S. lauensis* reconstructed using 559 bp-sequences revealed a newly identified species complex in the West Pacific BABs Lau, Futuna Volcanic Arc, North Fiji, Woodlark and Manus (Table 2, Fig 2A), supported by species delimitation results. Each basin contained one cryptic species of *S. lauensis*. The low $F_{st}$ values (0–0.15, p-values = <0.001) between the Fatu Kapa vent field population (within the Futuna Volcanic Arc) and the Lau Basin populations supported the hypothesis that specimens across the two basins are a single putative species and were thus grouped for downstream analyses (Table 3 in S4 File). All groups were monophyletic according to species delimitation conducted with ASAP, bPTP, and BDF (Fig 2A, Fig 4 in S2 File, Fig 2 in S3 File, Table 1 in S2 File). Despite the Bayesian and ML trees giving overall different topologies, due to the presence of a clock in the Bayesian tree and slight variation of evolutionary models within BEAST, the topology of the cryptic species clade and its relationship with *S. brevispina* was conserved across both trees. The ML values for all branches within the clade containing the cryptic species were considered sufficiently high (> 80%) to support the existence of cryptic species. However, the posterior probability value for the split between the Manus-Woodlark basins and the Lau-Futuna-Phoenix basins was 0.5. In spite of this, all shallower branches had high support values from both methods (Fig 2A, Fig 2 in S3 File). All three species delimitation methods, as well as high $F_{st}$ values (0.77, *p*-value = < 0.001) provided support for the presence of cryptic species.

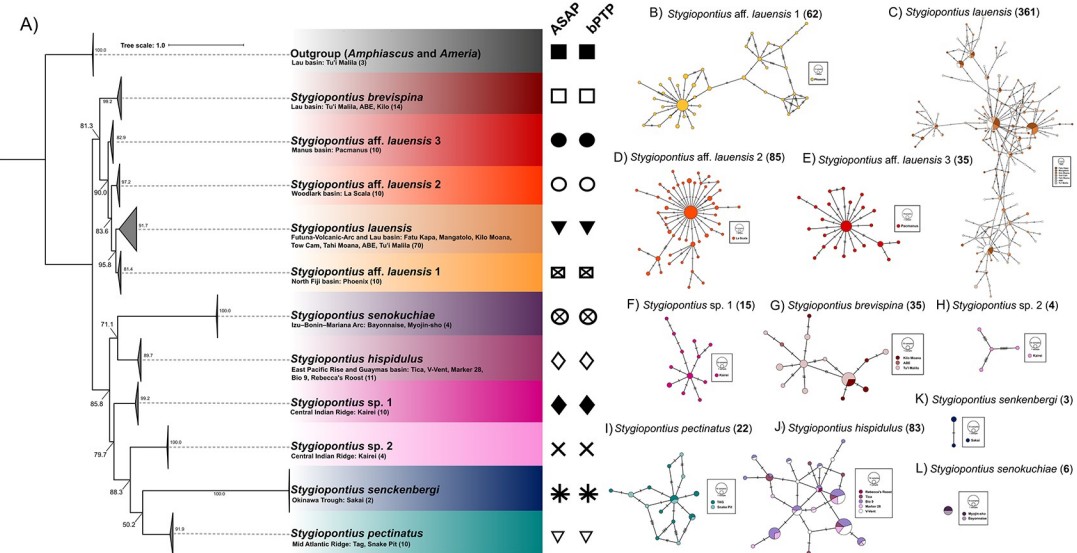

**Fig 2. Phylogenetic reconstruction of the vent endemic *Stygiopontius* genus and corresponding haplotype networks.** A) Maximum likelihood tree based on 559 bp of codon position partitioning of the mitochondrial *Cox1* for dirivultid copepods. Tree built with IQ-TREE (node values are ML bootstraps). Species are coloured according to their minimum spanning networks on the right: B) *Stygiopontius* aff. *lauensis* 1, C) *Stygiopontius lauensis*, D) *Stygiopontius* aff. *lauensis* 2, E) *Stygiopontius* aff. *lauensis* 3, F) *Stygiopontius* sp. 1, G) *Stygiopontius brevispina*, H) *Stygiopontius* sp. 2, I), *Stygiopontius pectinatus*, J) *Stygiopontius hispidulus*, K) *Stygiopontius senckenbergi*, and L) *Stygiopontius senokuchiae*. Networks are coloured on a gradient from north to south. Black symbols under ASAP and bPTP denote the species delimitation by both software (where matching symbols are found, congruence in the species delimitation is inferred by both software, and each symbol corresponds to one species.

**Molecular diversity.** Nucleotide diversity was generally very low across all *Stygiopontius* species, ranging from 0.001–0.01 with the lowest values exhibited in *S. senokuchiae* and the highest in *S. lauensis* (Table 3 in S4 File). Haplotype diversity was high across all of the *Stygiopontius* species, ranging from 0.8–1 with the lowest values belonging to populations of *S. brevispina* in the Lau Basin, and the highest values belonging to *S. lauensis*. All haplotypes were separated by one or two mutations with the exception of *Stygiopontius*. sp. 2 and *S. pectinatus* (Fig 2H and 2I). The $F_{st}$ values were variable across all species ranging from 0–0.12 (*p*-value = 0.01) for *S. senokuchiae* and *S. brevispina*, respectively. In the latter, over 12% of the variation occurred amongst the populations. The $F_{st}$ value for the of *S. lauensis* population was low 0.02 (*p*-value = < 0.001), with the majority of the variation occurring within the population (97.5%). These two species (*S. brevispina* and *S. lauensis* BAB species complex) form one half of the bifurcating phylogenetic tree and are the only species that exhibit structure (Fig 2A). In general, *S. lauensis* from the Lau and Futuna Volcanic Arc are well mixed with haplotypes shared between all localities and with most haplotypes separated by one or two mutations (Fig 2C). Haplotypes from *S*. aff. *lauensis* 1, 2, and 3, were restricted each to one basin and one site, but haplotypes were similarly connected by very few mutational steps (Fig 2B–2D). The $F_{st}$ values between *S. lauensis* and *S*. aff. *lauensis* 1, 2, and 3, were very high, ranging from 0.77–0.94 (*p*-value = <0.001) (Table 3). Da values between *S. lauensis* and *S*. aff. *lauensis* 1 (0.04–0.05) are half of that between *S. lauensis* and *S*. aff. *lauensis* 2 and 3 (0.07–0.97) and were low between populations of *S. lauensis* in the Lau basin ranging from -0.00035–0.002 (Table 3).

Minimum spanning networks for the *Stygiopontius* genus in Fig 2, revealed multiple network shapes across the species. *Stygiopontius* sp. 1, *S. brevispina*, and *S. hispidulus* exhibit "complex star"-shaped networks, typical of populations in which multiple high frequency

**Table 3. Differentiation and divergence between populations of the *S. lauensis* species complex.**

|  | Mangatolo | Fatu_Kapa | Tow_Cam | ABE | Tui_Malila | Tahi_Moana | Phoenix | La_Scala | Pacmanus | Kilo_Moana |
|---|---|---|---|---|---|---|---|---|---|---|
| **Mangatolo** | 0 | 0.00001 | 0.00025 | 0.00021 | 0.00016 | 0.00094 | 0.04601 | 0.08621 | 0.09378 | -0.0002 |
| **Fatu_Kapa** | -0.00015 | 0 | 0.00037 | 0.0003 | 0.00037 | 0.00123 | 0.04691 | 0.08772 | 0.09597 | 0.00006 |
| **Tow_Cam** | 0.00772 | 0.01526 | 0 | 0.00012 | 0.00072 | 0.00172 | 0.04416 | 0.09079 | 0.09714 | 0.00052 |
| **ABE** | 0.01225 | 0.01603 | 0.00311 | 0 | 0.00036 | 0.00104 | 0.04134 | 0.07713 | 0.0863 | 0.00045 |
| **Tui_Malila** | 0.01854* | 0.02616* | 0.04175* | 0.0269* | 0 | 0.00016 | 0.04048 | 0.07697 | 0.07848 | -0.00035 |
| **Tahi_Moana** | 0.06536* | 0.07922* | 0.09534* | 0.06024* | 0.00801 | 0 | 0.04216 | 0.0786 | 0.08242 | 0.00018 |
| **Phoenix** | 0.7919* | 0.78901* | 0.80012* | 0.77346* | 0.78138* | 0.79195* | 0 | 0.07204 | 0.08139 | 0.04044 |
| **La_Scala** | 0.89696* | 0.89531* | 0.90739* | 0.88919* | 0.88931* | 0.90386* | 0.91181* | 0 | 0.07369 | 0.0753 |
| **Pacmanus** | 0.89093* | 0.88952* | 0.89878* | 0.8756* | 0.88* | 0.89058* | 0.9186* | 0.93923* | 0 | 0.08027 |
| **Kilo_Moana** | -0.00437 | 0.01098 | 0.03765 | 0.02796 | -0.00365 | 0.00963 | 0.82111* | 0.93012* | 0.93229* | 0 |

Below the diagonal: Fst values (genetic differentiation) calculated from an AMOVA with 1000 permutations. Numbers with * indicate Fst values for which corresponding p-values were significant (alpha– 0.05). Above the diagonal: net divergence (Da) calculated using DnaSP6.

haplotypes exist, separated by very few mutations [81] (Fig 2F, 2G and 2J). In contrast, *Stygiopontius* sp. 2 exhibited a more complex haplotype relationship where the four haplotypes were separated by 4–6 mutation steps. *S. pectinatus* exhibited a "complex mutational" network, in which some branches are separated relatively more mutations than others [81] (Fig 2H). *S. lauensis* also exhibited a clear "complex star"-shaped network, where the multiple frequent haplotypes appear to be mostly represented by the Tu'i Malila, Mangatolo, and Fatu Kapa vent fields (Fig 2C). Meanwhile, *S.* aff. *lauensis* 1 exhibits a clear "complex mutational" network (Fig 2B). In contrast, *S.* aff *lauensis* 2 and 3 exhibited clear "star"-shaped networks, in which a single, widespread haplotype is typically positioned at the center of the network and is thought to be an ancestral haplotype [81] (Fig 2D and 2E). In this case additional haplotypes are linked to the dominant haplotype by very few mutational steps.

**Demographic history of the *Cox1* gene.** Demographic analysis revealed that most species of *Stygiopontius* are undergoing expansion to various degrees as implied by the negative Tajima's $D$ and Fu's $F_s$ values, as well low values of SSD and $rg$ with no significance (empirical data is not deviating significantly from the null model of sudden population expansion) (Fig 3, Table 4, Table 4 in S4 File). Tajima's $D$ for *S. lauensis* ranged from -1.53–0.38 and was overall significantly negative (Tajima's $D$ = -1.53, *p*-value = 0.03). Tajima's $D$ values were -1.93 (*p*-value = 0.01), -2.29 (*p*-value <0.001), and -2.27 (*p*-value = 0.004) for *S.* aff. *lauensis* 1, *S.* aff. *lauensis* 2, and *S.* aff. *lauensis* 3, respectively (Fig 3A–3D). Tajima's $D$ values for *S. brevispina* were -1.02 for specimens from Tu'i Malila and -1.48 for specimens from Kilo Moana (overall = -1.25, *p*-value = 0.13) (Fig 3E). For *S. hispidulus*, the values ranged from -0.27–0.95 (the highest of any species), with an overall value of 0.18 (*p*-value = 0.60) (Fig 3F). Tajima's $D$ values were -1.24 and -0.86 for populations of *S. pectinatus* in TAG and Snake-pit vent fields, respectively (Table 4 in S4 File). *Stygiopontius.* sp. 1 and *Stygiopontius.* sp. 2 had Tajima's $D$ values of -1.44 and -0.84, respectively (Fig 3G, Table 4 in S4 File).

Fu's $F_s$ is a more sensitive parameter to changes in population size and thus ranged broadly across the species. Values were -6.45 to -25.25 for *S. lauensis*, and -25.82, -26.79, -15.67 for *S.* aff. *lauensis* 1,2, and 3, respectively. All Fu's $F_s$ values were significantly negative for all four cryptic species (*p*-value <0.001). Combined with low SSD and $rg$ values and highly non-significant *p*-values for both parameters, *S. lauensis* and *S.* aff. *lauensis* 1 appear to be expanding populations (Fig 3A and 3B, Table 4, Table 4 in S4 File). However, *S.* aff. *lauensis* 2 and 3 both exhibit significant *p*-values for $rg$ (p-values = 0.01 and 0.04, respectively), suggesting that an

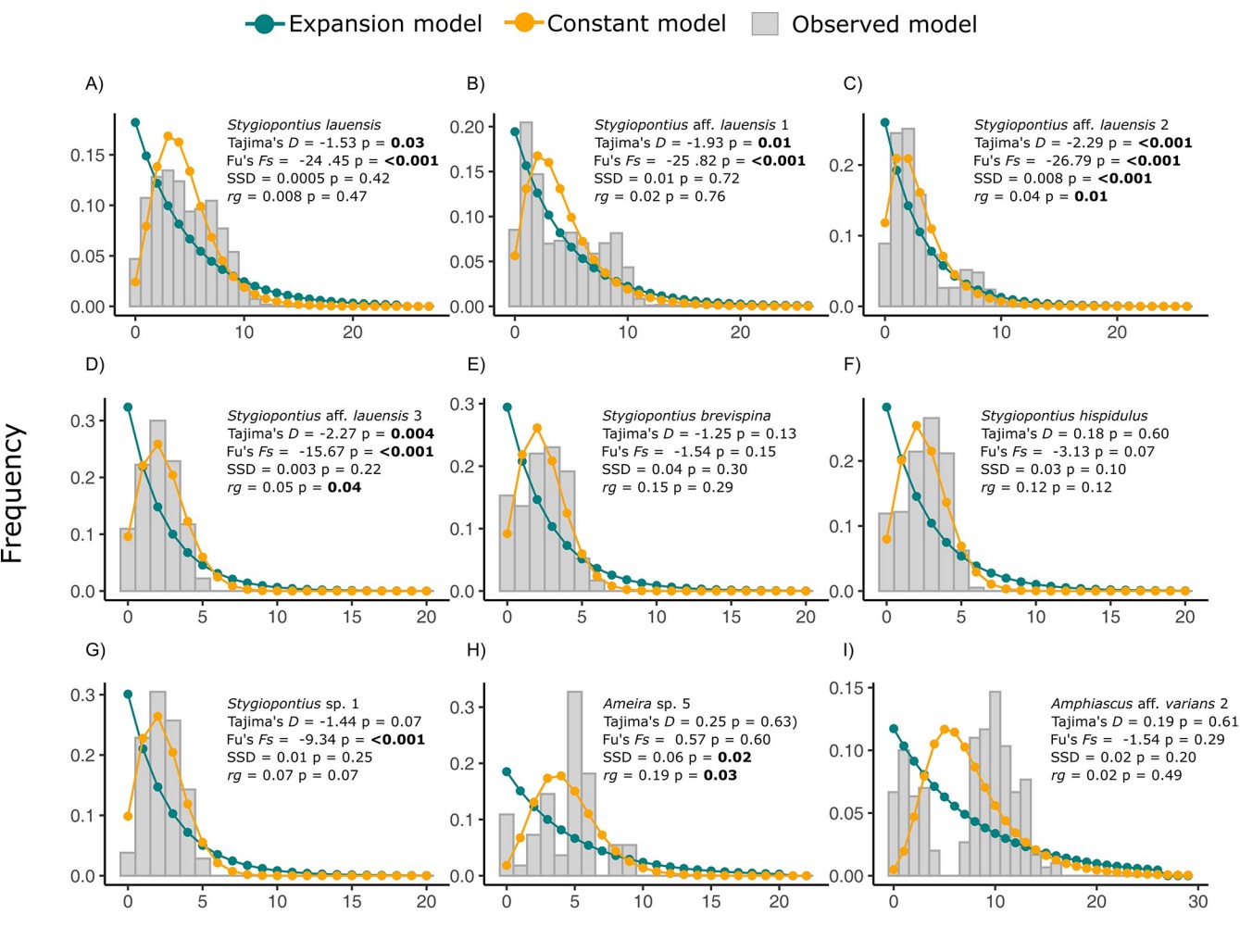

**Fig 3. Mismatch distribution plots.** a) *S. lauensis*: skewed multimodal, b) *S.* aff. *lauensis* 1: skewed multimodal, c) *S.* aff. *lauensis* 2: skewed bimodal, d) *S.* aff. *lauensis* 3: unimodal e) *S. brevispina*: bimodal f) *S. hispidulus*: unimodal, g) *Stygiopontius*. Sp.1: unimodal, h) *Ameira* sp. 4: multimodal, and i) *Amphiascus* aff. *varians* 2: bimodal. Grey bars represent the number of pairwise differences (or mismatches) between the sequences. The teal dotted line represents the distribution of mismatches under a model of sudden population expansion and the orange dotted line represents the number of mismatches under a model of constant population size.

alternative process (stability or selective pressures may be involved) (Fig 4C and 4D). Only the latter exhibited negative *rg* values out of all *Stygiopontius* species studied. For *S. brevispina*, Fu's Fs values were the highest of any species ranging from -1.19 to -1.9, with an overall value of -1.54 (*p*-value = 0.15). *S. hispidulus* exhibited slightly lower values, ranging from -0.8 to -4.85 (overall value = -3.13, *p*-value = 0.07), similar to those of *S. pectinatus* (-2.55 to -4.34), that had an overall value of -3.44 (*p*-value = 0.02). *S.* sp.1 and *S.* sp. 2 from the CIR exhibited quite different results with Fu's $F_s$ being lower in *S.* sp. 1 (-9.34, *p*-value <0.001) than in *S.* sp. 2 (-0.29, *p*-value = 0.26.) (Fig 4E–4G).

Bayesian skyline plots (Fig 4), calculated with a molecular clock of 0.3 mutations/Ma, also indicate population expansion in *S. lauensis* over the last 10 kya. In *S.* aff. *lauensis* 1, population growth occurred ~7.5 kya, and in *S.* aff. *lauensis* 2, and 3 population growth occurred ~1.5 kya. The largest increase occurred in *S. lauensis* with $N_e$ increasing 100-fold, followed by *S.* aff.

**Table 4. Summary information on population level dynamics for the main groups within the Southwest Pacific back-arc basins (BABs).**

| Population level dynamics | *S. lauensis* | *S.* aff. *lauensis* 1 | *S.* aff. *lauensis* 2 | *S.* aff. *lauensis* 3 | *Ameira* sp. 5 | *Amphiascus* aff. *varians* 2 |
|---|---|---|---|---|---|---|
| Basin | Lau-Futuna | North Fiji | Woodlark | Manus | Eastwall | Lau and Woodlark |
| Expanding/constant/bottleneck | Expanding | Expanding | Expanding | Expanding | Constant | Expanding |
| Time of population growth (kyr) | 10 | 7.5 | 1.5 | 1.5 | na | na |
| Degree of polymorphism | Low | Low | Low | Low | High | High |
| Tajima's *D* | Negative | Negative | Negative | Negative | Positive | Positive |
| Fu's $F_s$ | Negative | Negative | Negative | Negative | Positive | Negative |
| Significant *rg p*-value | No | No | Yes | Yes | Yes | No |
| Evidence of selective sweep | No | No | Maybe | Maybe | Maybe | Maybe |
| population structure | No | No | No | No | No | Yes |

Information for *S. lauensis*, *S.* aff. *lauensis* 1, *S.* aff. *lauensis* 2, *S.* aff. *lauensis* 3, *Ameira* sp. 5, and *Amphiascus*. Aff. *varians* 2. Information includes sampling location (basin), whether the population is undergoing expansion, stability or a bottleneck, the time in thousands of years to the population growth event, the degree of polymorphism, whether there is evidence of a selective sweep and whether there is population structure.

*lauensis* 1 and 2, and lastly 3, which appears to have experienced a very small increase in population size. All plots indicated that growth curves had either reached or were reaching a plateau and were becoming stable, explaining the multimodality in *S. lauensis* and *S.* aff. *lauensis* 1.

## Ameiridae

**Phylogenetic reconstruction and species delimitation.** Two clearly distinct clades for specimens belonging to the genus *Ameira* were identified, constituting *Ameira*. Sp. 4 from the Lau Basin, with haplotypes shared between Tu'i Malila and ABE (Fig 5A and 5B) and *Ameira*. Sp. 5 from Eastwall in the EPR (Fig 5A and 5C) which formed a clade with a group of North Sea *Ameira* nov. sp. Aff. *parvula* 1 and 2 [85].

**Molecular diversity.** *Ameira*. Sp. 4 exhibited high polymorphism (S = 17, out of six sequences), with higher levels of polymorphism in Tu'i Malila (S = 15) than in ABE (S = 5). Nucleotide diversity ($\pi$ = 0.01) and a *Hd* of 0.8 was found, with six specimens represented by four haplotypes and one shared haplotype between ABE and Tu'i Malila (Fig 5B). The $F_{st}$ values were 0 (p-value = 1) for this population, typically characteristic of a panmictic population, however, sample number was very low, and these results must be interpreted carefully. It is noteworthy that 100% of the variation occurred within populations for this species. Specimens of *Ameira*. sp. 5 were only identified from Eastwall, thus no population differentiation could be calculated. Within the population, 11 sequences were used containing 11 segregating sites. Nucleotide diversity was much lower than in *Ameira*. sp. 5 ($\pi$ = 0.007). A total of six haplotypes were found amongst 11 specimens with a *Hd* of 0.9 (Fig 5C). Genetic diversity for harpacticoid species were higher than for dirivultid species as calculated by the AMOVA (Table 1 in S4 File). Minimum spanning networks also revealed "complex mutational" networks in which some branches are separated by relatively more mutations than others.

**Demographic history of the *Cox1* gene.** Despite there being at least three sequences for the demographic calculations in many of the species, the number is still not sufficient for a

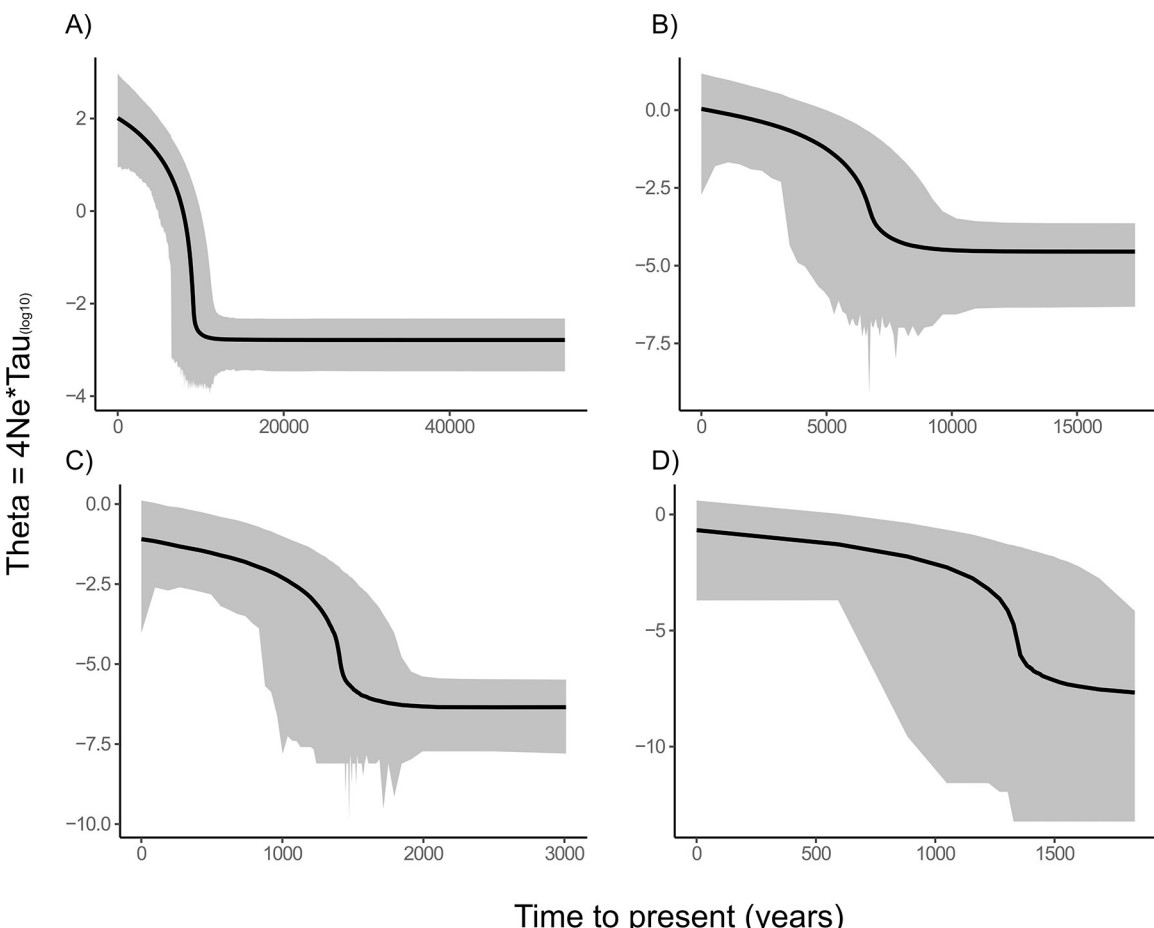

**Fig 4. Extended Bayesian skyline plots.** a) *Stygiopontius lauensis*, b) *Stygiopontius* aff. *lauensis* 1, c) *Stygiopontius* aff. *lauensis* 2, and d) *Stygiopontius* aff. *lauensis* 3. All models were calculated with a strict molecular clock of 0.3 for 200 million iterations to achieve adequate convergence. The Y axis was log transformed and the X axis represents time (years) to the present. The dashed black line represents the median and the straight black lines the 95% confidence intervals.

conclusive result of the demography of these organisms. Only species with at least ten sequences are discussed. In *Ameira*. sp. 5 from Eastwall, positive values were found for Tajima's *D* (0.25, *p*-value = 0.63) and Fu's $F_s$ (0.57, *p*-value = 0.60). SSD and *rg* were significantly high in this species pointing to stable population with values of 0.06 (*p*-value = 0.02) and 0.19 (*p*-value = 0.03), respectively (Table 4, Table 2 in S4 File). This was concurrent with the multimodal mismatch analysis (Fig 3H).

**Miraciidae.** *Phylogenetic reconstruction and species delimitation.* Two distinct clades were identified for specimens belonging to the genus *Amphiascus* namely *Amphiascus*. aff. *varians* 1 from the EPR and *Amphiascus*. aff. *varians* 2 from the West Pacific BABs (Fig 5A). Within each clade, high levels of divergence between individuals were encountered. The bPTP (Fig 3 in S2B File) showed high support (Bayesian posterior probabilities > 0.7) for cryptic species within the clade *Amphiacsus*. aff. *varians* 1. However, ASAP (Fig 1 in S2A File) and BFD showed support for the one species hypothesis (Table 1 in S2C File). Low $F_{st}$ values were also found within this group. *Amphiascus*. aff. *varians* 2 from the Lau and Woodlark basins was delimited to one genetic group by bPTP, ASAP, and BFD. Therefore, despite the very high $F_{st}$ values (0.72, *p*-value = <0.001), the group was retained as single species for further analysis. Specimens taxonomically identified as Diosaccina (Miraciidae) appeared as a sister species to

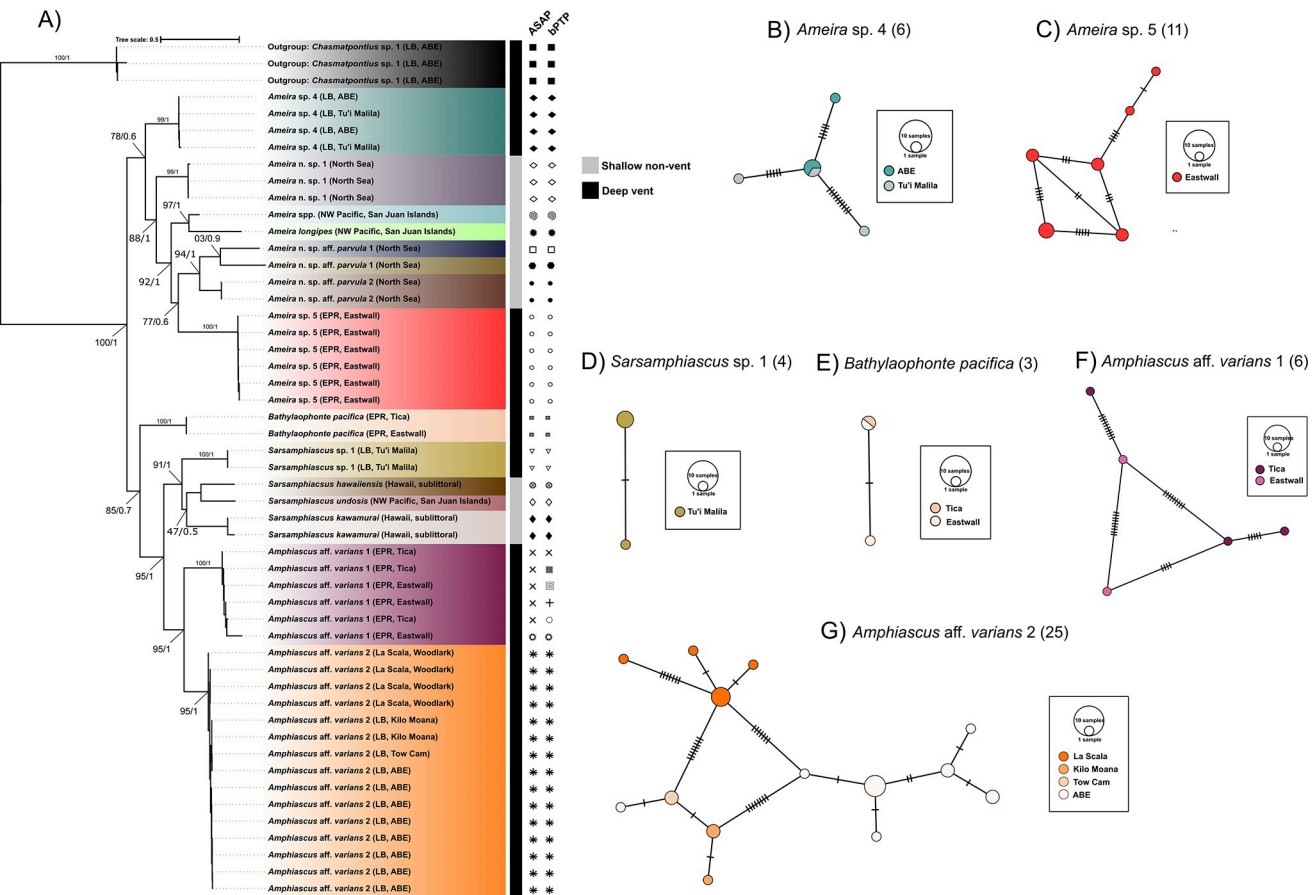

**Fig 5. Phylogenetic reconstruction of the harpacticoid families associated with vents and corresponding haplotype networks.** Maximum likelihood tree based on 544 bp of codon position partitioning of the mitochondrial *Cox1* for harpacticoid copepods. Trees built with IQ-TREE and BEAST (topologies of ML and BEAST trees were identical and so presented as a single tree). Node values are ML bootstrap/BPP. Species are colour-coded with their corresponding haplotype networks to the right of the phylogenetic tree: B) *Ameira* sp. 4, C) *Ameira* sp. 5, D) *Sarsamphiascus* sp. 1, E) *Bathylaophonte pacifica*, F) *Amphiascus* aff. *varians* 1, and G) *Amphiascus* aff. *varians* 2. Networks are coloured on a gradient from North to South. The black symbols under ASAP and bPTP denote the species delimitation by both software (where matching symbols are found, congruence in the species delimitation is inferred by both software, and each symbol corresponds to one species. The grey and black bar denotes whether the species came from shallow water non-vent environments or deep-water hydrothermal vents.

*Sarsamphiascus kawamurai* [86], from a sublittoral zone off the coast of Hawaii (Fig 5A), therefore, this species was designated as *Sarsamphiascus* sp. 1.

**Molecular diversity.** *Amphiascus*. aff. *varians* 1 represented by six sequences, exhibited considerably high levels of polymorphism (S = 69), with the majority of polymorphism occurring in Eastwall (S = 60, out of three sequences) (Table 1 in S4 File). Overall nucleotide diversity was the highest of any species in the study ($\pi$ = 0.07) and haplotype diversity was 1, indicating that most haplotypes are distantly related, as also observed in the haplotype network for this species, with some specimens separated by ten mutational steps (Fig 5F, Table 1 in S4 File). Despite this, $F_{st}$ values were low (0.01, *p*-value = 0.49) with 99.2% of the variation occurring within populations, indicating almost no structure within the species. However, the authors note that 6 sequences is not enough to draw conclusions about genetic structure and a large sample size combined with a broader sampling area would be required. *Amphiascus*. aff. *varians* 2, represented by 25 sequences and 27 polymorphic sites also exhibited low nucleotide diversity ($\pi$ = 0.01) and high $Hd$ (0.93). However, $F_{st}$ values were exceptionally high (0.72, *p*-value = <0.001) for this species, and the AMOVA results point to the majority of the variation

occurring between populations (72.4%) rather than within (27.6%), despite species delimitation methods suggestive of a monophyletic group, and haplotypes being separated by only eight mutational steps (Fig 5G). *Sarsamphiascus* sp. 1 was also only found within one site (Tu'i Malila) and therefore, no population genetics could be calculated. Within the four sequences representing the species, one polymorphic site was found and a nucleotide diversity $\pi$ of 0.5 was calculated. The four sequences were represented by two haplotypes ($Hd$ = 0.5) (Fig 5D). The minimum spanning network for *Amphiascus* aff. *varians* 1 showed a "complex mutational" network with some haplotypes separated by 4 mutations while others were separated by up to ten mutations. Meanwhile, *Amphiascus* aff. *varians* 2 exhibited what is described as a "reciprocally monophyletic" network, in which more than one lineage is present and each lineage is linked by a typically long branch associated with numerous mutations [81]. However, in this case, despite multiple lineages being visible, these were only separated by a maximum of eight mutations.

**Demographic history of the *Cox1* gene.** *Amphiascus* aff. *varians* 2 exhibited positive Tajima's *D* values (0.19, *p*-value = 0.61) and negative Fu's $F_s$ values (-1.54, *p*-value = 0.29) with no significant deviation from a sudden population expansion model (Table 2 in S4 File), in agreement with the mismatch analysis that matched well with a model of expansion (Fig 3I, Table 4). None of the remaining populations had sufficient sequences to confidently interpret parameters of demography.

**Laophontidae.** For *Bathylaophonte pacifica*, only three sequences were recovered (Table 1 in S4 File). One polymorphic site was found in the two sequences originating from Eastwall also with two haplotypes ($Hd$ = 0.67) with an overall nucleotide diversity $\pi$ of 0.001. Shared haplotypes were found between Eastwall and Tica (Fig 5E). Although the $F_{st}$ values were 0 (*p*-value = 1), the sample size was too small for conclusive results on population differentiation.

## Discussion

This study reports the presence of cryptic species within the highly abundant copepod species *Stygiopontius lauensis*. Populations are structured across the West Pacific BAB system from the Lau to Manus basins, with one cryptic species found per basin, namely *S. lauensis* a, b, c, and d, from the Lau, Futuna Volcanic Arc, North Fiji, Woodlark and Manus basins, respectively. This is in contrast to the *Stygiopontius* species from MORs, which exhibit high levels of similarity within and amongst vent sites and fields, as well as demonstrating population expansion. The results of this study are indicative of population structure occurring only in BABs, however, the sampling scale for the BABs was significantly larger than that for MORs in this study, and the authors recognise that had a large sampling scale been applied to the MORs investigated here, structure in the populations may have been encountered. Here, we also report the genetic diversity and demography of harpacticoid copepods from hydrothermal vent systems. Species of *Ameira* exhibited higher within-population structure than dirivultids found in BAB and/or MOR. Species of *Amphiascus* exhibited higher levels of structure within and between vent sites, with *Amphiascus*. aff. *varians* 2 from BAB exhibiting exceptionally high levels of populations structure, whilst being delimited as one species by species delimitation methods. Our results are indicative of heavily structured populations of *S. lauensis* and *Amphiascus*. aff. *varians* 2 in spite of their contrasting life history traits. We discuss our findings within the context of differing geological settings of BAB and MOR environments, as well as the different life-history traits of dirivultid and harpacticoid copepods.

### Geological features and population structure

Dirivultid copepods are vent endemic and exist only at depths between 700–4000 m with no records of dirivultid copepods from shallow hydrothermal vent fields [55]. BABs are enclosed geological features where basin walls act as prominent barriers to dispersal. Thus, depth-restricted organisms may become genetically isolated from organisms found in neighboring

basins. Within BABs, barriers to dispersal may be caused by regional currents, which, in the Lau basin, are known to have a mean northwest flow [87]. This could lead to asymmetric gene flow, resulting in a source-sink metapopulation system, promoting genetic differentiation by reducing gene diversity in sink populations, further exacerbated by hyper-local hydrodynamics [88]. It is worth noting that phylogeographic patterns differ when considering only the *Cox1* gene and when using genome-wide markers, and the latter suggest that dispersion is much more complex than previously estimated based on *Cox1* [89].

Radically different volcanic systems are formed in BABs in comparison to MORs. Interestingly, the results of this study indicate that a high level of genetic structure occurs only in the species belonging to BABs, namely *S. brevispina* and *S.* aff. *lauensis* 1, 2, and 3, an unprecedented finding for a genus that is otherwise well distributed across vent sites and fields along MORs [19,25,28,29]. *S. lauensis* is distributed across the Lau Basin over a distance of 775 km and exhibited the highest haplotype diversity values and very low nucleotide diversity, a typical trait of well-mixed populations that have experienced a bottleneck followed by a sudden population expansion [78] and is therefore, unlikely to be caused by small, long-term effective population sizes given their densities. No phenotypic differences were encountered between any of the specimens examined in the *S. lauensis* complex, indicative of recent allopatric speciation where morphological traits have yet to appear [90]. However, gene introgression via secondary contact events or the stabilising effect of homogenous vent conditions (i.e., temperature, hydrogen sulphide, oxygen, and heavy metal concentrations), may maintain morphological features such that they resemble the ancestral state, resulting in highly conserved morphological traits (morphological stasis) [91–93]. The number of mutational steps between *S. lauensis* and *S.* aff. *lauensis* 1 is only 12, indicative of recent divergence. Meanwhile, the number of mutations between *S. lauensis* and *S.* aff. *lauensis* 2, was 24 and between *S. lauensis* and *S.* aff. *lauensis* 3, was 28 (Fig 2B–2D). Similarly, Da values are approximately twice as large between *S. lauensis* and *S.* aff. *lauensis* 1 than between *S.* aff. *lauensis* 2 and 3, suggesting that divergence took twice as long for the latter two allopatric forms under a constant molecular clock [81].

In contrast to *S. lauensis*, *S. brevispina* exhibits exceptional genetic structure between proximate vent fields within the Lau Basin ($F_{st}$ = 0.1 between Tu'i Malila and Kilo Moana). Recent studies within the Lau Basin, have discovered that seafloor spreading is far more complex than previously anticipated [94]. Spreading rates may vary between Tu'i Malila (48 mm/year$^{-1}$) and vents to the north of ABE and Tahi Moana (120 mm/year$^{-1}$) [95], giving rise to very different local venting and as such different environmental conditions and nutritional sources that *S. brevispina* may have locally adapted to. A growing body of literature is emerging detailing dirivultid copepods as feeding predominantly on chemoautotrophs [8,34,40–42,96]. These studies elucidate two important findings, firstly that the gut contents of dirivultid copepods are mostly composed of chemoautotrophic bacteria based on lipid composition and depleted $\delta^{13}C$ signatures [41], and secondly, that there is a gradient of specialisation across species in the family, where some members are found very close to the vent source (e.g., *S. quadrispinosus*) and have evolved to contain high levels of hemoglobin to cope with depleted oxygen levels [23,97]. With increasing distance from the vent, (where nutritional sources may be more heterogenous [42]) further specialisation include a morphology better suited to feeding on bacterial mats at vents, whereby the morphology of mouthparts differs in the mouth aperture and bacteria-detecting plumose setules are present, an adaptation which is characteristic of *Benthoxynus scupilifer*.

## Life history, habitat use and community structure

Dirivultid copepods have only a few eggs, disperse via lecithotrophic nauplii, are endemic to deep-sea hydrothermal vents, and occur in high densities. In contrast, the three harpacticoid

families discussed here have numerous eggs, and can disperse via bottom or pelagic currents, are not restricted to the vent environment but also occur in the proximate non-venting areas, and contain genera shared between shallow and deep-water environments [20,21,26,27]. Despite these different life-traits, similar genetic patterns are observed for the three harpacticoid families.

**Dirivultidae.** Dirivultid copepod nauplii are lecithotrophic, relying on a single egg yolk for food during dispersal, and as such, they may not be able to disperse far. However, [19] estimated a generation turnover of 33 generations/year for dirivultid copepods living in environments of 20˚C which would translate into a larval dispersal duration (LDD) of 11 days, and 10 generations/year for those living in environments of 10˚C, resulting in a LDD of 36.5 days. Although LDD for many marine organisms represents only a fraction of the generation time, it is highly likely that LDD approaches the generation time for vent copepods, given that they only settle out of the water column once having developed adequate crawling legs, and reports of specimens at vents that had very recently reached adulthood [26]. Following this trend, in ambient deep-water temperatures of ~2˚C this would be even longer (3 generations/year, LDD = 121.6 days), a theory evaluated in [98], who reported that population connectivity and effective population size should, in general, be inversely related to ocean temperature. Further, [39] observed the nauplii of the dirivultid *S. pectinatus* hatching on board from collections of 80–300 m above vents in the MAR. They suggested that these nauplii might be carried away from vents in a dormant state, developing and settling only once an appropriate vent environment can be detected. Many marine invertebrate larvae can delay metamorphosis in the absence of suitable environmental cues conducive to settlement [99], thereby increasing larval lifespan and associated LDD. This phenomenon might allow larvae/nauplii to drift with currents for prolonged periods of time, possibly explaining the lack of genetic structure in dirivultid copepods from MORs and within the Lau Basin.

Demographic parameters, Tajima's $D$, Fu's $F_s$, SSD and $rg$, (Fig 3, Table 4 in S4 File) indicate that populations in the Lau basin are undergoing expansion, typical of populations exhibiting significant neutrality statistics and "star" or "complex"-star-shaped networks. *S.* aff. *lauensis* 1 also appears to be undergoing expansion at least for part of the population, as suggested by the significant neutrality tests and the "complex mutational" network, in which half of the network appeared to be represented by a dominant haplotype presumed to be an ancestral form, while the other half of the network was represented by multiple haplotypes mostly separated by three or four mutations. Populations belonging to the Woodlark and Manus basins (*S.* aff. *lauensis* 2 and 3) exhibit clear "star"-shaped networks indicative of populations undergoing sudden expansion despite the significant $rg$ values, that typically indicate a departure from sudden expansion. This was supported by the good fit of a model representing sudden expansion to the empirical data in the mismatch distribution analysis. Mismatch distributions from a population undergoing balancing selection, can also produce an increase of haplotypes at intermediate frequencies that may counter-balance the effect of an expansion. Conversely, a selective sweep is likely to exacerbate and/or mimic a population expansion [81], due to hitchhiking variant sites being inherited along with the beneficial mutation. This process promotes genetic homogeneity in the entire region surrounding the mutation and genome-wide markers would therefore be needed to resolve these findings. Finally, *S.* aff. *lauensis* 2, and *S. brevispina* appear to be bimodal, likewise indicative of constant populations, but may also be a result of mixing lineages, particularly if the bimodality is very expressed.

The EBSP analysis (Fig 4) suggests a logarithmic increase in the effective population size (EPS) of *S. lauensis* and *S.* aff. *lauensis* 1, 2, and 3. Interestingly, a quantifiable increase in EPS was noted for *S. lauensis* in the Lau Basin ~10 kya, whereas for *S.* aff. *lauensis* 1 (North Fiji), this occurred ~7.5 kya. This increase in EPS occurs even later for *S.* aff. *lauensis* 2 and 3 at ~

1.5 kya, which may coincide with a surge in volcanic eruptions across the Papua New Guinea and Tonga-Kermadec arcs, with an increase from 10 to 50% of global eruptions occurring in the last two thousand years [100]. This surge in the frequency of volcanic eruptions throughout the region would give rise to a higher density of vents, thus allowing vent-endemic copepods to colonise more vents via stepping stone vents [9]. The latter study postulates that phantom hydrothermal vents and intermediate populations are the reason why populations of *Bathymodiolus* mussels exhibit elevated levels of genetic connectivity between BABs and the Indian Ridge, and this has also been demonstrated for other vent-associated megafaunal gastropods and crustaceans on a basin-scale [101]. A similar scenario may therefore be applicable to dirivultid copepods.

Increases in effective population size may also be aided by constant reproduction, a phenomenon that is theorized for dirivultid copepods [26]. The latter study reported mating pairs of *S. quadrispinosus* at the Juan de Fuca Ridge that were all copepodite stage V and that females were oviferous as soon as they were adult, indicating that these copepods begin mating almost as soon as they settle. However, this behaviour may also be attributed to precopulatory mate guarding, also known to occur in harpacticoid copepods [102]. Dirivultid copepods are highly gregarious, existing in high densities as adults at vents [20,21,24,25,29,33,40]. In addition, as discovered by [18], nauplii and copepodite I stages are highly abundant in the plumes above the vents (50/1000 L water, 3 m above a vent). The ratio of males to females for *S. lauensis* is considerably higher than that found for other species of *Stygiopontius* [21,26,103], where often only males or female were reported (due to apparent gender niche partitioning in species belonging to this genus). This increase in the number of males further supports the possibility of constant reproduction. Furthermore, high levels of self-recruitment may be expected for dirivultid copepods, as depicted for vent fauna in general [104].

**Ameiridae, Miraciidae, and Bathylaophontidae.** Dispersal of harpacticoids via their planktotrophic nauplii may only happen occasionally by bottom or pelagic currents, as the nauplii from the harpacticoid families in this study are typically benthic-dwelling [47]. Highly structured populations of *Amphiascus* from the Lau to the Woodlark Basin indeed point to other factors than frequent long-distance dispersal driving genetic differentiation between metapopulations. Connectivity via planktotrophic nauplii, can make dispersal vulnerable due to disruptions in the form of strong current regimes, patchy food supplies, predation, and fluctuations in environmental conditions before settlement [56,105]. Importantly, climate change is already exacerbating these fluctuations directly impacting phytoplankton abundance [106], with consequences for organic matter that benthic harpacticoid copepods feed on and thus, may push vulnerable larvae and nauplii beyond their tolerance to increasing temperatures, salinity, and lower oxygen levels, further affecting their ability to colonize appropriate environments. Mortality rates may therefore be high in harpacticoid copepods that disperse via bottom and pelagic currents, and benthic dwelling may be the dominant form of recruitment.

The increased number of eggs generally found in harpacticoid copepods relative to dirivultid copepods may be a survival strategy to cope with environmental stressors. Tajima's $D$ and Fu's $F_s$ were both positive for *Ameira*. Sp. 5 indicative of high frequencies of shared or common polymorphism (Fig 3I, Table 2 in S4 File). The calculated SSD and $rg$ indices were both significant, indicating a strong deviation from a model of sudden population expansion. These results are typical of a stable population undergoing balancing selection, but also of a recent or contemporary, mild bottleneck [81]. The mismatch analysis for the *Ameira*. sp. 5 showed a multimodal distribution also associated with constant population size, results which were indicative of population stability (Fig 3I). Harpacticoid copepods are typically shallow-water families and genera (species from the North Sea, San Juan Islands, and the Hawaiian littoral zone) and it is therefore likely that only few haplotypes have colonized the deep-sea from

shallow waters resulting in bottleneck or founder effect signatures in the *Cox1* gene. However, more sequences would have to be analysed to draw conclusive results.

In contrast, demographic parameters of Tajima's *D* and Fu's $F_s$ for the Miraciidae were generally low with only Fu's $F_s$ being negative (Table 2 in S4 File) indicative of the onset of population expansion. This is supported by the low and not-significant SSD and *rg*, an interesting result for a population with such high $F_{st}$ values. One interpretation is that a significant and ongoing allele introgression of the *Cox1* gene is at play, resulting in contemporary structure but ongoing homogenisation of the gene. The mismatch analysis points to two mixing lineages (Fig 3I), and good model fit to the empirical data, supporting the theory that this is an expanding population. Thus, self-recruitment via the benthic-dwelling nauplii may play a role in maintaining EPS in harpacticoid copepods associated with the vent environment, as this phenomenon is not exclusive to vent-endemic species and is exhibited by many benthic taxa in the deep-sea [107].

The few sequences reported in this study are not enough to draw conclusions about diversity, phylogeny, or demography for *Bathylaophonte*, however taxonomy-based studies have found species belonging to the *Bathylaophonte* genus from vents around the Azores and Easter Island [108,109]. [33] have reported *B. pacifica* from the EPR, [21] have reported *B. pacifica* from the Lau Basin, and more recently [110] have reported members of the Laophontidae from the vents on Myojin and Bayonnaise Knolls and the Myojin-sho caldera. However, these regional identifications require subsequent confirmation using molecular methods. This family and in particular the *Bathylaophonte* genus are clearly widespread yet there is a paucity of data for this family and genus to date. Overall, the differences in population connectivity of the *Cox1* gene between different harpacticoid families associated with hydrothermal vents appear to be highly variable and may reflect their resilience to a changing ocean and/or variable environmental conditions at vents.

Harpacticoids are generalists, thus they can inhabit both basalt and sediment substrates. At the ABE hydrothermal field, they exist only where oxygen concentrations are high [21]. However, in the Okinawa Trough, they were found at both ends of the environmental gradient [40], suggestive of a tolerance to a broader range of environmental conditions than previously thought. The $\delta^{15}N$ and $\delta^{13}C$ ratios measured from harpacticoids resembled surface sediments but also varied highly, indicating little discrimination in nutritional sources for Ameiridae, Miraciidae and Bathylaophontidae [40]. These results indicate that abundance of food rather than food type may be more important in structuring harpacticoid communities at vents.

### Implications for area-based management tools and environmental management plans

Some of the studied vent fields within the Manus Basin such as the Solwara Nautilus Minerals prospects, or Tu'i Malila and ABE within the Lau Basin, are explored for their metal resources and may be subject to future anthropogenic disturbance in the form of deep-sea mining of SMS [11,110]. The Southwest Pacific BAB system harbors cryptic species of abundant vent-endemic copepods that may not be able to disperse across the basins, or may disperse very infrequently, reflected in their limited geographical distribution with very little genetic connectivity. This implies that populations in adjacent basins may not have the ability to recolonize and repopulate a vent if deep-sea mining should occur, collectively posing a risk of extinction of meta-populations on an inter-basin scale.

Biophysical models of larval dispersal may be used to inform environmental management on population connectivity [111,112]. However, currently the vast majority of the data informing these models only incorporate data from macro- and megafaunal communities at vents.

Recent studies focusing on the latter have elucidated higher levels of complexity in connectivity patterns with the likelihood of the emergence of cryptic species of megafauna increasing in BAB systems compared to MORs based on *Cox1* and genome-wide markers [101].

The co-occurrence of megafauna and meiofauna in deep-sea hydrothermal settings is poorly understood, yet they do co-occur in all deep-sea hydrothermal vents [8,24,84,113]. This co-occurrence means that the risk of biodiversity loss in ecosystems designated for deep-sea mining is likely higher when all size classes are considered, as opposed to studies only considering macro- and megafaunal communities, which collectively only consider the resilience of ~ 50% of the animal communities. Since these smaller size classes make up the majority of metazoan biodiversity in those systems, we strongly recommend the inclusion of meiofauna for such models in future investigations. To address this knowledge gap, ongoing studies (Diaz-Recio Lorenzo et al., *in prep*) are now incorporating the smaller size classes in the growing list of studies on population genomics focusing on the vent environment.

Trait-based approaches are fast growing in deep-sea research with much of the diversity studies focused on functional differences between groups of organisms. This approach has proved to be more beneficial than using species-based or operational taxonomic unit-based approaches alone, as it can be used to compare across species, phyla, and even entire ecosystems, providing a "common currency" with which to evaluate system processes and identify systems that are the most vulnerable to disturbance [14]. The function of highly abundant vent-endemic copepods within vent systems in the Southwest Pacific is poorly understood, however, studies so far point to a potentially important role in the chemoautotrophic food web of hydrothermal vents [34,40–42]. The function of sparse populations of harpacticoids at vents is currently unknown, as is the role of the food-rich vent for these harpacticoids that are not restricted to the vents. Our results on copepod population structure can help to better evaluate any potential future harm to copepod populations that may arise from mining of metal resources in the Southwest Pacific BAB system.

## Conclusion

Two main groups of copepods, the Siphonostomatoida (Dirivultidae) and Harpacticoida (Ameiridae, Miraciidae, and Bathylaophontidae)were studied to investigate the effects of life history traits on diversity and connectivity based on mtDNA. Results point to a higher likelihood of cryptic species and population structure occurring in back-arc basins (BABs) than on MORs for vent-endemic copepods. Despite potential for widespread harpacticoid species across BABs via planktotrophic nauplii and their lack of dependence on the vent environment, high levels of population structure also occur, likely as a result of predominant self-recruitment via benthic-dwelling nauplii and spatial heterogeneity of nutritional sources, suggesting that different life history traits can lead to similar patterns of genetic connectivity and structure. With regards to demography however, a dependence on the vent environment may have facilitated massive population expansion in dirivultid copepods, whereas the lack of that dependence in the three families of harpacticoid copepods detailed in this study might have promoted population stability (with the exception of the Miraciidae).

Together with vent obligate dirivultids, vent facultative harpacticoids contribute to the most abundant higher animal taxon in vent ecosystems, the Copepoda. Higher sampling effort is needed with regards to harpacticoid copepods across different vent settings to obtain a clearer picture of population structure and demographic history of the *Cox1* gene. However, mitochondrial DNA exhibits a fraction of the true structure and demography of populations and future research should include application of genome-wide markers to improve species delimitation and accuracy in determining fine scale structure and demographic patterns in

vent copepods. Such information can help evaluate risks of species extinction due to potential future anthropogenic impacts such as deep-sea mining and can inform regional environmental management plans, safeguarding biodiversity.

## Supporting information

**S1 File. Sequence metadata.**
(XLSX)

**S2 File. Species delimitation.**
(DOCX)

**S3 File. Bayesian posterior probability parameters.**
(DOCX)

**S4 File. Parameters of diversity and demography for all species.**
(DOCX)

## Acknowledgments

We warmly thank Charles R. Fisher, chief scientist, for the invitation to the 2009 TN235 cruise and sharing samples, Roxanne Beinart, chief scientist, for the donation of the samples collected during the 2016 FK160407 cruise, and the crew and scientific members of the 2019 Chubacarc cruise, who collected and sorted the copepod samples on board L'Atalante and kindly donated them. We are also grateful to Nadinne Jeanneth van den Hooven-Iza for help with the DNA extractions.

## Author Contributions

**Conceptualization:** Didier Jollivet, Sabine Gollner.

**Data curation:** Coral Diaz-Recio Lorenzo.

**Formal analysis:** Coral Diaz-Recio Lorenzo, Tasnim Patel.

**Funding acquisition:** Sabine Gollner.

**Investigation:** Coral Diaz-Recio Lorenzo, Eve-Julie Arsenault-Pernet, Camille Poitrimol, Pedro Martinez Arbizu.

**Methodology:** Coral Diaz-Recio Lorenzo.

**Project administration:** Sabine Gollner.

**Resources:** Didier Jollivet, Pedro Martinez Arbizu, Sabine Gollner.

**Supervision:** Didier Jollivet, Sabine Gollner.

**Validation:** Didier Jollivet, Sabine Gollner.

**Visualization:** Coral Diaz-Recio Lorenzo.

**Writing – original draft:** Coral Diaz-Recio Lorenzo.

**Writing – review & editing:** Tasnim Patel, Eve-Julie Arsenault-Pernet, Camille Poitrimol, Didier Jollivet, Pedro Martinez Arbizu, Sabine Gollner.

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
