## [Decision Letter · Decision Letter 0]

23 Jun 2023

PONE-D-23-15095Highly structured populations of deep-sea copepods associated with hydrothermal vents across the Southwest Pacific, despite contrasting life history traits.PLOS ONE

Dear Dr. Diaz-Recio Lorenzo,

Thank you for submitting your manuscript to PLOS ONE. After careful consideration, we feel that it has merit but does not fully meet PLOS ONE’s publication criteria as it currently stands. Therefore, we invite you to submit a revised version of the manuscript that addresses the points raised during the review process.

We look forward to receiving your revised manuscript.

Kind regards,

Hans G. Dam, Ph. D.

Academic Editor

PLOS ONE

Journal Requirements:

4. We note that Figure 1 in your submission contain map/satellite images which may be copyrighted. All PLOS content is published under the Creative Commons Attribution License (CC BY 4.0), which means that the manuscript, images, and Supporting Information files will be freely available online, and any third party is permitted to access, download, copy, distribute, and use these materials in any way, even commercially, with proper attribution. For these reasons, we cannot publish previously copyrighted maps or satellite images created using proprietary data, such as Google software (Google Maps, Street View, and Earth). For more information, see our copyright guidelines: http://journals.plos.org/plosone/s/licenses-and-copyright.

Additional Editor Comments:

Your manuscript was evaluated by three experts. Their recommendations varied from minor revision (2) to reject (1).  The reviewer recommending rejection thinks your study has fatal flaws. Because of the conflicting recommendations, I think a major revision is warranted unless you can convince all of us that the reviewer recommending rejection has fundamentally misunderstood your paper. I am therefore, offering you the possibility of a resubmission with a detailed rebuttal that addresses all reviewers' concerns and suggestions. If you do resubmit, I will ask all three reviewers for their views on how you have addressed all questions and concerns, not just their own. 

Reviewers' comments:

Reviewer's Responses to Questions

**Comments to the Author**

1. Is the manuscript technically sound, and do the data support the conclusions?

Reviewer #1: Yes

Reviewer #2: Yes

Reviewer #3: No

2. Has the statistical analysis been performed appropriately and rigorously? 

Reviewer #1: Yes

Reviewer #2: Yes

Reviewer #3: No

3. Have the authors made all data underlying the findings in their manuscript fully available?

Reviewer #1: Yes

Reviewer #2: Yes

Reviewer #3: No

4. Is the manuscript presented in an intelligible fashion and written in standard English?

Reviewer #1: Yes

Reviewer #2: Yes

Reviewer #3: Yes

5. Review Comments to the Author

Reviewer #1: This manuscript is a study of phylogenetic structure and life history traits of hydrothermal vent-associated copepods, one of the most abundant and diverse members of vent communities which is nonetheless very poorly known. It includes inter- and intraspecific phylogenetic structure based on CO1 data from 14 copepod species (plus 4 newly identified cryptic species) spanning 4 copepod families collecting comprising 708 specimens from 25 sites and corresponding trait data for each family. It is a valuable contribution to the scientific literature both in terms of copepod phylogeny and population structure and is also of broader interest to deep sea biology and hydrothermal vent communities.

I have number of minor suggestions:

1. I assume the authors plan to do this, but all new sequences should be uploaded to NCBI.

2. A table should be provided in the supplemental information that includes the taxonomic information, collection metadata, and NCBI numbers for all the new sequences.

3. I did not see the figure captions for the figures in the main text. These should be checked carefully. Especially for the phylogenetic tree figures, it is critical that the tree is reported in the captions (i.e., which tree is displayed, the highest likelihood tree from ML analysis of the consensus tree from BI analysis) and how the support values are given (support values appear to be BS/PP in some figures or just PP in others).

4. Throughout the manuscript the genus name for Ameira and Amphiascus should be written out in full, not abbreviated to A. in sections of the text where both taxa are referred to since it can be difficult for the reader to follow which genus is being discussed. (e.g., page 14)

5. When used at the start of a sentence, the full genus name should be written out, not abbreviated (e.g., A. sp. 5 on page 26 and A. aff. varians 1 on page 27).

6. When a shortened taxon name is used (e.g., dirivultid or harpacticoid) it should not be capitalized unless it is at the start of a sentence (e.g., page 32).

7. Standardize usage of “cox1” vs “Cox1” throughout the manuscript. I would suggest simply “CO1” although any of the above are acceptable, just be consistent.

8. Remove the following lines form the Phylogenetic Reconstruction section of Methods since this is more of an explanation of common phylogenetic knowledge and not a detail of the methods.

“Codon-usage bias can result in different mutation rates for different coding regions of a gene and therefore, different evolutionary histories for each coding region which, if not modelled correctly can lead to erroneous phylogenies and interpretation of species delimitation methods [61-65]. Therefore…”

Some details are missing from the Phylogenetic Reconstruction Methods section and it should be expanded to include:

9. Details of ML analysis: what tree is shown presented? The maximum likelihood tree or the majority rule consensus (the highest likelihood tree should be presented).

10. Details of the BI analysis (a) number of chains (b) number of generations (c) frequency of sampling (d) amount of burn-in generations (e) how convergence between chains was assessed (visually with tracer or statistically?).

11. Support value information. How many BS replicates in ML analysis?

12. Species delimitation methods: For BPTP wow many MCMC generations were completed (should be 100,000 at the minimum) and what was the sampling frequency and burn-in?

13. I recommended rewording the following sentence on page 21 to switch the word order from “The other Stygiopontius species had been morphologically previously identified by…” to “The other Stygiopontius species had been morphologically identified previously by…”

14. In some cases, the phrase “exist only at depth” (e.g., page 33, 35) is used. Specify what you mean by “at depth”. At depths over 1,000m for example?

15. Page 38 put “e.g., S. quadrispinosus” in parentheses.

16. Page 43, Conclusions. To emphaise which 2 groups of copepods you are referring to, I would reorganize how the taxa are mentioned from:

“Two main groups of copepods, the Dirivultidae (Siphonostomatoida) and the Ameiridae, Miraciidae, and Bathylaophontidae (Harpacticoida)…”

to:

“Two main groups of copepods, the Siphonostomatoida (Dirivultidae) and Harpacticoida (Ameiridae, Miraciidae, and Bathylaophontidae)…”

17. Page 44. Consider rewording the following sentence, it sounds a bit awkward: “…to accurately determine population genetic parameters of structure and demography”

18. Page 44. “Can help evaluating” should be changed to “Can help evaluate” or “Can help with evaluating”.

Reviewer #2: Review of Lorenzo et al. PlosOne

Lorenzo et al. did a very thorough study of deep-sea copepods population structure and demography across the Southwest Pacific using the mitochondrial cytochrome oxidase I gene. After identifying specimens morphologically, they performed a phylogenetic analysis and discover cryptic species in Stygiopontius lauensis. They then further their sampling in the back-arc basins for phylogeographical and demographical analyses. They found population structure within species and evidence of population expansion hypothesized being linked to an increase of hydrothermal vent activities after the Last Glacial Maximum.

This paper is very long and could have been split into multiple papers (either per copepod family, or phylogeny versus phylogeography). However, because of the comparative aspect of the study, I understand the authors’ desire to incorporate the complete study into a single comprehensive paper. The paper would nevertheless benefit from some reorganizations that would make it an easier flow, but these re-organizations should be easy to achieve.

I recommend accepted the paper after minor revisions. Below are my comments and edits.

Introduction

Line 69 – “and as well as”

Line 74 – add references to the papers

Line 160-163- should refer to Fig. 1 to help the reader

Materials and Methods

Line 174 – “kapa” to “Kapa”

Line 176 – I would refer to Fig 1 and table 2 together here, so readers can find site names more easily.

Line 177 – “along the EPR” is a bit misleading since all sampling sites are from 9N. Replace “along the East Pacific Rise” by “from the East Pacific Rise”

Line 179 – Spell out “Izu–Bonin–Mariana (IBM)”

Unfortunately, the line numbers were not provided after this point

Page 14 – Missing closing parenthesis “refer to [18, 19, 57, 58]” to “refer to [18, 19, 57, 58])”

Page 15 – Explain in the methods why only the females were dissected. It’s not clear at this point why the dissection is needed for the study. Based on the result section, I understand the female have specific “characters that discriminate” them from other species. It would help knowing this when reading the methods.

Page 15 - “pictures taken with the Leica Application suite”, however pictures were not provided with the paper. Either provide the picture (supplementary material) or remove the end of the sentence as it is not needed for the study.

Page 15 – please add PCR mix conditions, especially since you are using 2 sets of primers. Or indicate you followed the PCR mix conditions from [59] if that is the case.

Page 16 – which primer pairs were used to sequence?

Page 16 – The way the cleaning of sequences is written is confusing. If ambiguities were resolved appropriately using the chromatograms, then there should not be any gaps (insertion/deletion), nor frame shift. If, as stated, some sequences had to be removed because of frame shift and stop codon found after cleaning the sequences makes me wonder if the all the used cleaned sequences were of good enough qualities or if some mutation may be the result of sequences that have not been sufficiently verified using the chromatograms. If chromatograms were not used in the initial cleaning, then authors should verify them to ensure all sequences used in further analysis are indeed unambiguous.

Page 16 – “blasted against the NCBI database […] allowed to differ by 20%” Which blast was used? What parameter was used with regards to the 20%? The e-value?

Page 18 – “Further, ….. between two models” incomplete sentence

Page 19 – Need to make it clear analyses of diversity and demography are being done among populations of the same species or species complex

Page 19 – “DnaSP was used to first define populations”  based on vent sites? Clarify

Page 20 – “To further investigate (…) Cox1 gene. Extended” replace the dot by a coma

Page 20 – “[82] and were constructed” missing information or remove “and”

Results

Comments:

While the order of the method section should remain the same, I found myself lost in the result section. The result section would be better re-organized per family as suggested below. It would make for an easier flow to fully understand an entire family before moving onto the next family. For example, while reading the molecular diversity and looking at haplotype networks in the Dirivultidae, I found myself wanting to know the demography of that family, which doesn’t come until much later in the paper.

As is, the results are difficult to follow, especially because figures and tables are sometime out of order. Figures/tables, including supplementary material, should be renumbered to follow the order they are cited in the text.

The authors need to provide the accession numbers of the sequences.

Suggested order for the result section:

1. Dirivultidae

1.1. Morphological identification of the S. lauensis species complex

1.2. Phylogenetic reconstruction and species delimitation

1.3. Molecular diversity (including haplotype networks)

1.4. Demography

2. Ameiridae

2.1. Phylogenetic reconstruction and species delimitation

2.2. Molecular diversity (including haplotype networks)

2.3. Demography

3. Miraciidae

3.1. Phylogenetic reconstruction and species delimitation

3.2. Molecular diversity (including haplotype networks)

3.3. Demography

Edits:

Page 20 – Results: italicize Stygiopontius lauensis

Page 21 – Change to: The phylogeny of S. lauensis reconstructed using X bp-sequences revealed…

Page 21 and through – Stay consistent in the entire manuscript “Table” or “table”

Page 22 – After the end of the Dirivultidae paragraph, please describe the network shown in Fig 4 and 5.

Page 22 - A sp.4 and A sp5 are inversed in the text. A sp. 5 on the figure is the one from East Wall and A sp.4 from ABE.

Page 22 – The figures numbering if off. Fig 6, (not Fig 5) has Ameira. Also, there is no mention in the text of Fig 4 and 5 before

Page 24: italicize Stygiopontius

Page 24 – Period after (Fig 4c and d).

Page 27 – 6 sequences for A. aff varians 1 is small to conclude there is “no structure within the species”. Also, both sites are from 9°N. You may detect structure within the species if you had samples from the entire EPR. I would rephrase that you did not detect structure, but higher sampling size and broader sampling area are needed.

Discussion

Comment:

Page 32 – “This is in contrast to Stygiopontius from MORs, …. population structure occurring only in BABs”, and page 33 “only in the species belonging to BABs”, and page 43-44 “Results point to … than on MORs for vent-endemic copepods”

The authors made strong statements of population structure being much stronger in BAB than in MORs. I wouldn’t make such strong statements as it may just be due to differences in sampling scales in the study. There is a very different geographical scale between the sampling area in the BABs (>17° of latitude from Manus to Lau) versus the MORs. In Stygiopontius hispidulus, the authors only had samples from 9°N (EPR) and a single sequence from GC. Had they sampled along the entire EPR, they may have found a greater diversity and greater population structure in Stygiopontius hispidulus they were not able to capture with their current samples. This is especially the case as 9°N and GC are both north of the equator, and south EPR is a different biogeographical region. Similarly on MAR, TAG and Snake Pit are only 3° of latitude apart.

Edits:

Page 33 – “this could to asymmetric gene flow”  missing verb

Page 37 – Add “(LGM)” after Last Glacial Maximum

Through the paper: be consistent in using either UK English or US English. For example, specialization/specialisation (page 38), among/amongst,

Page 38 – “Dirivultids are vent-specialists … Benthoxinus scupilifer”. This paragraph does not feel tight to the study. A sentence is needed to explain why being vent specialists, and the specialization gradient is important in the context of this genetic study and the discussed effective population size (or would that paragraph be better moved after discussing the population structure?).

Page 41 “in in”

Tables and Figures

Figures/tables, including supplementary material, should be renumbered to follow the order they are cited in the text.

Tables 2, 3 and 4

Because of formatting, I was not able to see the entire tables. Tables was cut-off.

Fig 1.

Add “Futuna Volcanic Arc” on the 1.b map.

It would help readers who are unfamiliar with deep-sea sites if authors could link the sampling sites on the map to the ones listed in Table 2. For example, authors could add a superscript number by each vent field and add that superscript number next to the vent field in Table 2.

North Sea1, Hawaii2, etc

Fig. 4, 5 and 6

Re-order the sites in the colored legends based on geographical location (north to south). Keep the same order in the figure legend description.

Fig 7

Legend: There is no “red dotted line”. You have yellow (constant) and blue/green (expansion).

Fig 8

Legend: “dashed black line” and “straight black line” do not correspond to the figure  change to “black line” and “shaded grey area”.

Reviewer #3: I congratulate the great effort put into sampling and identifying copepods from the deep sea, especially in hydrothermal vents. However, the value for delimiting cryptic species, the lack of input parameters for the models, the statistical significance of the analyses, and the citations of molecular clocks lead me to reject this work. I believe that by modifying and explicitly addressing these issues, you will be able to publish it in another journal without any problems. However, since these aspects were not specified in this review, your analyses are not reproducible. Best of luck and encouragement!

6. PLOS authors have the option to publish the peer review history of their article (what does this mean?). If published, this will include your full peer review and any attached files.

Reviewer #1: No

Reviewer #2: No

Reviewer #3: No

---

## [Author Response · Author response to Decision Letter 0]

29 Aug 2023

Manuscript number: PLOS ONE: PONE-D-23-15095 - [EMID:633287e6e72cd20b]

Highly structured populations of deep-sea copepods associated with hydrothermal vents across the Southwest Pacific, despite contrasting life history traits.

Dear Dr. Diaz-Recio Lorenzo,

Thank you for submitting your manuscript to PLOS ONE. After careful consideration, we feel that it has merit but does not fully meet PLOS ONE’s publication criteria as it currently stands. Therefore, we invite you to submit a revised version of the manuscript that addresses the points raised during the review process.

We look forward to receiving your revised manuscript.

Kind regards,

Hans G. Dam, Ph. D.

Academic Editor

PLOS ONE

Additional Editor Comments:

Your manuscript was evaluated by three experts. Their recommendations varied from minor revision (2) to reject (1). The reviewer recommending rejection thinks your study has fatal flaws. Because of the conflicting recommendations, I think a major revision is warranted unless you can convince all of us that the reviewer recommending rejection has fundamentally misunderstood your paper. I am therefore, offering you the possibility of a resubmission with a detailed rebuttal that addresses all reviewers' concerns and suggestions. If you do resubmit, I will ask all three reviewers for their views on how you have addressed all questions and concerns, not just their own.

Response: 

Dear editor, 

We would like to thank you and all of the reviewers for taking the time to read our manuscript and give feedback, and for considering this manuscript for publication in PLOS ONE. We would like to re-submit this manuscript having revised the document and documents pertaining to it with major revisions applied, as requested. We have addressed the minor comments as suggested by reviewer 1 and 2. Reviewer 3 had some major concerns especially about the methods used, which we clarify in the revised version. The misunderstanding stems mainly from too little detail provided in the methods section in the original manuscript from our side, and we are confident that this misunderstanding is now solved by our additional explanations we give especially in the methods. In addition, we would like to note that all maps were made from publicly available data and constructed using open source QGIS software by the corresponding author and this has now been stated in the manuscript and all figures have been run through the PACE image corrector software, thus meeting the PLOSONE requirements. We would also like to state that there was no discrepancy between the financial statements, but this was suggested during review.

Please see here below our detailed response.

Reviewer #1:

This manuscript is a study of phylogenetic structure and life history traits of hydrothermal vent-associated copepods, one of the most abundant and diverse members of vent communities which is nonetheless very poorly known. It includes inter- and intraspecific phylogenetic structure based on CO1 data from 14 copepod species (plus 4 newly identified cryptic species) spanning 4 copepod families collecting comprising 708 specimens from 25 sites and corresponding trait data for each family. It is a valuable contribution to the scientific literature both in terms of copepod phylogeny and population structure and is also of broader interest to deep sea biology and hydrothermal vent communities.

I have number of minor suggestions:

1. I assume the authors plan to do this, but all new sequences should be uploaded to NCBI.

Response: Firstly, we thank the reviewer for the constructive comments. With regard to the data uploaded on NCBI. All data is available on NCBI using the accessions provided in the data availability statement during submission. The data was set to be available from May 31st 2023 and should therefore have been publicly available during the revision process, but the authors apologise if there was a technical issue. All accessions in the data availability statement are now accessible on NCBI. 

2. A table should be provided in the supplemental information that includes the taxonomic information, collection metadata, and NCBI numbers for all the new sequences.

Response: We agree with the reviewer and a table including taxonomic information, collection metadata, and NCBI numbers have been added as a table in the first supplementary material file S1 in excel form. 

3. I did not see the figure captions for the figures in the main text. These should be checked carefully. Especially for the phylogenetic tree figures, it is critical that the tree is reported in the captions (i.e., which tree is displayed, the highest likelihood tree from ML analysis of the consensus tree from BI analysis) and how the support values are given (support values appear to be BS/PP in some figures or just PP in others).

Response: All figure captions were included and visible in the submitted document. The reviewer here refers to particularly the phylogenetic tree figures, where captions are visible in the submitted document, and already include information requested by the reviewer such as which tree is displayed and how the support values are given (Page 23, line 452). With regard to the fact that support values appear to be in BS/PP in some figures or just PP in others, this is because for the Dirivultidae we present two different trees as the two different tree building methods gave different topologies. Therefore, for the Dirivultidae, one tree contains only BS values and the other contains PP values. However, the caption states that values are in BS/PP when it should state that the values are BS for the ML tree and PP for the Bayesian tree. This has now been corrected. 

4. Throughout the manuscript the genus name for Ameira and Amphiascus should be written out in full, not abbreviated to A. in sections of the text where both taxa are referred to since it can be difficult for the reader to follow which genus is being discussed. (e.g., page 14)

Response: The authors agree that this is not easy to follow and have changed all instances of “A.” to their corresponding, unabbreviated names. 

5. When used at the start of a sentence, the full genus name should be written out, not abbreviated (e.g., A. sp. 5 on page 26 and A. aff. varians 1 on page 27).

Response: The authors agree, and all instances of this have been changed. See response to comment 4 pertaining to this. 

6. When a shortened taxon name is used (e.g., dirivultid or harpacticoid) it should not be capitalized unless it is at the start of a sentence (e.g., page 32). 

Response: The authors agree and have changed all instances of capitalised shortened names throughout, unless they appear at the start of a new paragraph, or after a full stop. 

7. Standardize usage of “cox1” vs “Cox1” throughout the manuscript. I would suggest simply “CO1” although any of the above are acceptable, just be consistent.

Response: The authors agree and have changed all instances of this error have been changed. 

8. Remove the following lines form the Phylogenetic Reconstruction section of Methods since this is more of an explanation of common phylogenetic knowledge and not a detail of the methods.

“Codon-usage bias can result in different mutation rates for different coding regions of a gene and therefore, different evolutionary histories for each coding region which, if not modelled correctly can lead to erroneous phylogenies and interpretation of species delimitation methods [61-65]. Therefore…”

Response: The authors think it is important to justify this in the text, but it can be condensed. Therefore, the text above has been replaced with the following: 

“Alignments were partitioned into coding frames 1, 2, and 3, to account for codon-usage bias, which if not modelled correctly can lead to erroneous phylogenies [61-65].” Please refer to page 17, lines 299-300. 

Some details are missing from the Phylogenetic Reconstruction Methods section and it should be expanded to include:

9. Details of ML analysis: what tree is shown presented? The maximum likelihood tree or the majority rule consensus (the highest likelihood tree should be presented).

10. Details of the BI analysis (a) number of chains (b) number of generations (c) frequency of sampling (d) amount of burn-in generations (e) how convergence between chains was assessed (visually with tracer or statistically?).

11. Support value information. How many BS replicates in ML analysis?

Response: The authors agree these parameters should have been clearly stated in the methods. These have now been added on page 17 from lines 306-316. 

“Duplicate sequences were removed prior to tree construction to avoid inflation of support values. Trees were run with 1000 bootstraps and only the highest likelihood tree is displayed. Bayesian posterior probabilities were calculated using unlinked site models, unlinked clock rates, and linked trees (Fig 1 in S3), allowing different evolutionary rates for each partition. Site models were found separately using the Beast Model Test and applied automatically to each partition alongside a strict clock of 1.0. Parameters of the multispecies coalescent were 1.0 for the population mean and linear with constant root for the population function (ploidy was set to mitochondrial). For the priors, the tree construction model was the Yule model, and all standard parameters were retained. A chain length of 200 million, sampling frequency 5000, and burnin of 10 % was used. “

12. Species delimitation methods: For BPTP wow many MCMC generations were completed (should be 100,000 at the minimum) and what was the sampling frequency and burn-in?

Response: The authors agree these parameters should have been clearly stated in the methods. Standard values of the bPTP webserver were used: 

https://species.h-its.org/

These include a burnin of 10 %, thinning of 100, and a MCMC chain of 500,000. This information has now been added to page 18, lines 334-338.

“Secondly, the Bayesian Poisson Tree Process (bPTP) [68] was used, a process that models branching events in a phylogenetic tree in terms of number of substitutions. This was done using both the ML trees and the BPP trees produced from the alignments in Newick format, and the default parameters from the PTP webserver phylogenetic tree (MCMC chain = 500,000, thinning =100, and burnin = 10 %).”

13. I recommended rewording the following sentence on page 21 to switch the word order from “The other Stygiopontius species had been morphologically previously identified by…” to “The other Stygiopontius species had been morphologically identified previously by…”

Response: The authors now refer the editor and reviewers to page 22, lines 424-425 and agree that this reads better. The sentence has been changed to the suggested text above. 

14. In some cases, the phrase “exist only at depth” (e.g., page 33, 35) is used. Specify what you mean by “at depth”. At depths over 1,000m for example?

Response: The first instance of this ( now on page 39, line 813) has been modified to include the recorded depth range (700-4000). The second instance of this (now on page 41, line 859) has been removed as the authors thought this was too repetitive and therefore redundant. 

15. Page 38 put “e.g., S. quadrispinosus” in parentheses.

Response: This refers now to page 45, line 949. The text has been placed in parentheses. 

16. Page 43, Conclusions. To emphaise which 2 groups of copepods you are referring to, I would reorganize how the taxa are mentioned from:

“Two main groups of copepods, the Dirivultidae (Siphonostomatoida) and the Ameiridae, Miraciidae, and Bathylaophontidae (Harpacticoida)…”

to:

“Two main groups of copepods, the Siphonostomatoida (Dirivultidae) and Harpacticoida (Ameiridae, Miraciidae, and Bathylaophontidae)…”

Response: This refers now to page 50 lines 1069-1070. The authors agree this reads better and have changed the text to that suggested above. 

17. Page 44. Consider rewording the following sentence, it sounds a bit awkward: “…to accurately determine population genetic parameters of structure and demography”

Response: The authors refer the editor and reviewers to page 51, lines 1085-1090. We agree this is a bit awkward sounding and have changed it to the following: 

“Higher sampling effort is needed with regards to harpacticoid copepods across different vent settings to obtain a clearer picture of population structure and demographic history of the Cox1 gene.”

18. Page 44. “Can help evaluating” should be changed to “Can help evaluate” or “Can help with evaluating”.

Response: This grammatical mistake on page 51, line 1092, has been corrected. 

Reviewer #2:

Lorenzo et al. did a very thorough study of deep-sea copepods population structure and demography across the Southwest Pacific using the mitochondrial cytochrome oxidase I gene. After identifying specimens morphologically, they performed a phylogenetic analysis and discover cryptic species in Stygiopontius lauensis. They then further their sampling in the back-arc basins for phylogeographical and demographical analyses. They found population structure within species and evidence of population expansion hypothesized being linked to an increase of hydrothermal vent activities after the Last Glacial Maximum. 

This paper is very long and could have been split into multiple papers (either per copepod family, or phylogeny versus phylogeography). However, because of the comparative aspect of the study, I understand the authors’ desire to incorporate the complete study into a single comprehensive paper. The paper would nevertheless benefit from some reorganizations that would make it an easier flow, but these re-organizations should be easy to achieve. 

I recommend accepted the paper after minor revisions. Below are my comments and edits.

Introduction

Line 69 – “and as well as”

Response: Firstly, we would like to thank the reviewer for their constructive comments. With regard to the comment above, the “and” has been removed.

Line 74 – add references to the papers

Response: The authors refer the editor and reviewers to pages 3-4, lines 71-76. And assume that the reviewer refers to the papers pertaining to the megafauna. These references are added in the next sentence when the topics of the papers about the aforementioned megafauna are elucidated. However, the authors have turned this into one sentence to make sure it is clear that the papers refer to that same megafauna and to avoid repetition. 

Line 160-163- should refer to Fig. 1 to help the reader

Response: This can now be found on page 9, line 163-164. The authors agree and have referred to Fig. 1b to help the reader understand the biogeography of the Stygiopontius species complex. 

Materials and Methods

Line 174 – “kapa” to “Kapa”

Response: The authors have corrected this on page 10, line 175.

Line 176 – I would refer to Fig 1 and table 2 together here, so readers can find site names more easily. 

Response: The authors assume the reviewer means Fig. 1a and b, and have added this to the text to help readers find the site names on page 9, line 177.

Line 177 – “along the EPR” is a bit misleading since all sampling sites are from 9N. Replace “along the East Pacific Rise” by “from the East Pacific Rise”

Response: The authors agree with this and have changed “along” to “from”, on page 10, line 178. 

Line 179 – Spell out “Izu–Bonin–Mariana (IBM)”

Response: This is indeed the first instance and the full name should have been written. This has now been corrected on page 9, line 180.

Unfortunately, the line numbers were not provided after this point

Response: The authors apologise for this error and the inconvenience it may have caused during revision. All line numbers are now inserted and will be referred to throughout the response to reviewers. 

Page 14 – Missing closing parenthesis “refer to [18, 19, 57, 58]” to “refer to [18, 19, 57, 58])”

Response: The closing parenthesis has been added on page 13, line 208.

Page 15 – Explain in the methods why only the females were dissected. It’s not clear at this point why the dissection is needed for the study. Based on the result section, I understand the female have specific “characters that discriminate” them from other species. It would help knowing this when reading the methods. 

Response: Only females of Dirivultidae copepods were used for this study. This is because they are considerably larger than males and yield more DNA than the males. With regard to why a dissection was conducted, morphological differences can only be found during dissection as they may be extremely small. Those small differences give way to new species and had that been the case, these would not be cryptic species but rather new species that would need taxonomic descriptions. Clarification has been added to the text on page 14, line 233-234.

Page 15 - “pictures taken with the Leica Application suite”, however pictures were not provided with the paper. Either provide the picture (supplementary material) or remove the end of the sentence as it is not needed for the study. 

Response: This sentence has been removed on page 14, lines 236-237. 

Page 15 – please add PCR mix conditions, especially since you are using 2 sets of primers. Or indicate you followed the PCR mix conditions from [59] if that is the case.

Response: These are not two sets of primers. They are one set of primers that have M13 tails attached to them (this is how they are ordered from the company, included in the standard list of primers from Macrogen), that aid in sequencing. However, the authors recognise that this was picked up by two of the reviewers and have modified the text to make this clear on page 15, 256-261.

“The Cytochrome c. oxidase subunit I gene (Cox1) was amplified using the Folmer [59] primer sets LCO1490 (5’GGTCAACAAATCATAAAGATATTGG’3), and CO2198 (5’TAAACTTCAGGGTGACCAAAAAATCA’3). These primers were ordered with corresponding M13 tails attached: M13F-pUC (-40) (5’GTTTTCCCAGTCACGAC’3), and M13R-pUC (-40) (5’CAGGAAACAGCTATGAC’3), for LCO1490 and CO2198, respectively.”

Page 16 – which primer pairs were used to sequence? 

Response: Please see response to previous comment to resolve this question. 

Page 16 – The way the cleaning of sequences is written is confusing. If ambiguities were resolved appropriately using the chromatograms, then there should not be any gaps (insertion/deletion), nor frame shift. If, as stated, some sequences had to be removed because of frame shift and stop codon found after cleaning the sequences makes me wonder if the all the used cleaned sequences were of good enough qualities or if some mutation may be the result of sequences that have not been sufficiently verified using the chromatograms. If chromatograms were not used in the initial cleaning, then authors should verify them to ensure all sequences used in further analysis are indeed unambiguous.

Response: The authors agree that this is confusing. All sequences were verified using chromatograms in Geneious Prime. Only sequences that were too short were removed from the analysis. There were no sequences that produced frame shifts or stop codons after cleaning and therefore the text is misleading but this was indeed checked for by using a translation table for mitochondrial DNA. As there were no stop codons, the alignment was deemed suitable for downstream analysis. This has been rewritten for clarification (pages 16, lines 278-292). 

“Cleaned contigs from the S. lauensis species were batch-blasted against the NCBI nucleotide collection and optimised for highly similar sequences using the Megablast option. Any sequences not pertaining to Dirivultidae were removed, representing contamination, resulting in an amplification success rate of 77 %. The remaining sequences had E-values of 0 and matched with S. lauensis as the first hit. Sequences were aligned using the MAFFT (version 7.490) plugin in Geneious [60] and cleaned manually within the Geneious interface. The subsequent alignment was translated into protein sequences using the translation table for invertebrate mitochondria, allowing reading frame identification of each nucleotide sequence separately to check for stop codons following methods from [19]. As no stop codons were found, the alignment was deemed suitable for downstream analysis.”

Page 16 – “blasted against the NCBI database […] allowed to differ by 20%” Which blast was used? What parameter was used with regards to the 20%? The e-value? 

Response: The sequences were blasted using NCBI’s blast against the nucleotide collection and optimised using Megablast for highly similar sequences. The fact that is says “were allowed to differ by 20 %” in the text is erroneous and refers to the fact that the alignment contained sequences that had a % similarity of 80 % (i.e. 20 % dissimilar). The sequences were simply blasted and any results that did not have a hit with a dirivultid copepod were discarded. All the rest of the sequences came back as S. lauensis, E-values were either 0 or negligible in each instance and in each instance, S. lauensis matched as the first hit. The authors have clarified this in the text and show an example below of where the “20 %” dissimilarity came from and have modified the text to clarify this section. Please see page 16, lines 279-285.

Page 18 – “Further, ….. between two models” incomplete sentence

Response: This sentence has been completed, please refer to pages 18, line 340.

Page 19 – Need to make it clear analyses of diversity and demography are being done among populations of the same species or species complex

Response: The authors state on page 19, line 358-359, that putative species alignments were used for diversity and demographic analyses. A few words have been added here to make this clearer. 

“To perform pairwise, permutational analyses of molecular variance (AMOVA), DnaSP v.6 [70] was used to first define populations the vent field that each specimen was collected from within each putative species alignment.”

Page 19 – “DnaSP was used to first define populations” � based on vent sites? Clarify

Response: The authors recognised that this should have said specimens and not “populations”. The populations are already defined as their putative species, and for the AMOVA, we delimit within each population, the specimens by sampling location or vent field. Please see, as for the previous comment, page 19, line 358-359.

Page 20 – “To further investigate (…) Cox1 gene. Extended” replace the dot by a coma

Response: This has been corrected on page 20, line 383.

Page 20 – “[82] and were constructed” missing information or remove “and”

Response: Please see comment above pertaining to this sentence. 

Results

Comments:

While the order of the method section should remain the same, I found myself lost in the result section. The result section would be better re-organized per family as suggested below. It would make for an easier flow to fully understand an entire family before moving onto the next family. For example, while reading the molecular diversity and looking at haplotype networks in the Dirivultidae, I found myself wanting to know the demography of that family, which doesn’t come until much later in the paper.

As is, the results are difficult to follow, especially because figures and tables are sometime out of order. Figures/tables, including supplementary material, should be renumbered to follow the order they are cited in the text. 

Response: The authors generally agree with this comment and have changed the order of the results section to match the suggestion by the reviewer. 

The authors need to provide the accession numbers of the sequences.

Response: All accession numbers are available in the data availability statement and are freely available on NCBI. However, the authors agree that a table containing metadata, taxonomy, and accession number of each individual should be added to the supplementary. This has now been done in the form of a new S1 containing an excel sheet with metadata pertaining to all sequences used in the study including taxonomy, region, vent field, associated megafauna, sampling equipment, source, sequence ID, genbank accession. Extra metadata was not included as this can be found in table 2 (e.g. coordinates). 

Suggested order for the result section:

1. Dirivultidae

1.1. Morphological identification of the S. lauensis species complex

1.2. Phylogenetic reconstruction and species delimitation

1.3. Molecular diversity (including haplotype networks)

1.4. Demography

2. Ameiridae

2.1. Phylogenetic reconstruction and species delimitation

2.2. Molecular diversity (including haplotype networks)

2.3. Demography

3. Miraciidae

3.1. Phylogenetic reconstruction and species delimitation

3.2. Molecular diversity (including haplotype networks)

3.3. Demography

Response: The authors agree this is a more coherent order of information and have changed it to match the order. 

Edits:

Page 20 – Results: italicize Stygiopontius lauensis

Response: The authors have gone through all documents using Ctrl F to make sure that all instances of species name are correctly italicised. 

Page 21 – Change to: The phylogeny of S. lauensis reconstructed using X bp-sequences revealed… 

Response: This has been changed, please see page 22, line 432. 

Page 21 and through – Stay consistent in the entire manuscript “Table” or “table” 

Response: The authors have gone through the documents using Ctrl F to make sure that all instances of “Table” and “table” are homogenised. 

Page 22 – After the end of the Dirivultidae paragraph, please describe the network shown in Fig 4 and 5.

Response: The authors agree this has not been discussed properly, however all figures have been condensed to include phylogeny and networks and all networks are now described and discussed. 

Page 22 - A sp.4 and A sp5 are inversed in the text. A sp. 5 on the figure is the one from East Wall and A sp.4 from ABE. 

Response: This has been corrected. Please see page 31, lines 617-626. 

Page 22 – The figures numbering if off. Fig 6, (not Fig 5) has Ameira. Also, there is no mention in the text of Fig 4 and 5 before 

Response: The authors have added a paragraph describing Figures 4 and 5 as requested by the reviewer above. The instance where Fig 5 is written instead of Fig 6 has been corrected and the entire order of Figs in general throughout the newly structured results section has been revised. 

Page 24: italicize Stygiopontius

Response: All instances where species names are mentioned have been correctly formatted to italicise the Latin names. 

Page 24 – Period after (Fig 4c and d).

Response: All figures have been reordered and grammar corrected throughout pertaining to these. 

Page 27 – 6 sequences for A. aff varians 1 is small to conclude there is “no structure within the species”. Also, both sites are from 9°N. You may detect structure within the species if you had samples from the entire EPR. I would rephrase that you did not detect structure, but higher sampling size and broader sampling area are needed.

Response: The authors agree, and this has been added to page 34, lines 707-709.

“However, the authors note that 6 sequences is not enough to draw conclusions about genetic structure and a large sample size combined with a broader sampling area would be required.”

Discussion

Comment:

Page 32 – “This is in contrast to Stygiopontius from MORs, …. population structure occurring only in BABs”, and page 33 “only in the species belonging to BABs”, and page 43-44 “Results point to … than on MORs for vent-endemic copepods”

The authors made strong statements of population structure being much stronger in BAB than in MORs. I wouldn’t make such strong statements as it may just be due to differences in sampling scales in the study. There is a very different geographical scale between the sampling area in the BABs (>17° of latitude from Manus to Lau) versus the MORs. In Stygiopontius hispidulus, the authors only had samples from 9°N (EPR) and a single sequence from GC. Had they sampled along the entire EPR, they may have found a greater diversity and greater population structure in Stygiopontius hispidulus they were not able to capture with their current samples. This is especially the case as 9°N and GC are both north of the equator, and south EPR is a different biogeographical region. Similarly on MAR, TAG and Snake Pit are only 3° of latitude apart.

Response: The authors agree with this comment, and have addressed this in the text to incorporate the uncertainty in the sampling scale. Please see pages 39, lines 817-821.

“The results of this study are indicative of population structure occurring only in BABs, however, the sampling scale for the BABs was significantly larger than that for MORs in this study, and the authors recognise that had a large sampling scale been applied to the MORs investigated here, structure in the populations may have been encountered.”

Edits:

Page 33 – “this could to asymmetric gene flow” � missing verb 

Response: The word “lead” has been added to page 40, line 819. 

Page 37 – Add “(LGM)” after Last Glacial Maximum

Response: The authors have removed references to the LGM as it is not strictly relevant to the study and is not discussed in detail. Instead we refer to 10 kya. 

Through the paper: be consistent in using either UK English or US English. For example, specialization/specialisation (page 38), among/amongst, 

Response: The authors have reviewed any instances of US English and changed it to UK English. 

Page 38 – “Dirivultids are vent-specialists … Benthoxynus scupilifer”. This paragraph feels untight to the study. A sentence is needed to explain why being vent specialists, and the specialization gradient is important in the context of this genetic study and the discussed effective population size (or would that paragraph be better moved after discussing the population structure?).

Response: The authors have added this paragraph to discuss the adaptations of dirivultids to the vent environment and their relationship with chemoautotrophic bacteria as further in the text, we compare this to the harpacticoids who have significantly more heterogenous nutritional sources. The authors agree that this paragraph fits better under the text discussing population structure and has now been moved. 

Page 41 “in in”

Page 48, line 1015

Reviewer #3:

I congratulate the great effort put into sampling and identifying copepods from the deep sea, especially in hydrothermal vents. However, the value for delimiting cryptic species, the lack of input parameters for the models, the statistical significance of the analyses, and the citations of molecular clocks lead me to reject this work. I believe that by modifying and explicitly addressing these issues, you will be able to publish it in another journal without any problems. However, since these aspects were not specified in this review, your analyses are not reproducible. Best of luck and encouragement!

Response: The authors would like to thank reviewer three for taking the time to critically review our manuscript. The misunderstanding stems mainly from too little detail provided in the methods section in the original manuscript from our side, and we are confident that this misunderstanding is now solved by our additional explanations we give especially in the methods. 

Abstrat:

Line 41: I find the selected sites that do not have any hydrothermal effect to be biased, as there are 22 sites with an effect and only 3 without it.

Response: The authors would like to point out that this is inherently not the point of the study. We are not comparing deep-sea vent copepods to shallow water species in any quantitative or statistical way, to find out how the effect of hydrothermal activity affects diversity. The use of shallow water species was singularly to elucidate the phylogenetic positioning of deep-sea harpacticoid copepods relative to their shallow water shared genera. 

Introduction:

Line 77: I don't understand what is meant by "50% biodiversity" in relation to what.

Response: The authors agree that this was unclear and have added the word “metazoan” in front of biodiversity, to highlight that we refer strictly to animals. The meiofauna accounts for > 50 % of the metazoan biodiversity at hydrothermal vents. Please see page 4, line 77. 

Line 88: It discusses the abundance of copepods in hydrothermal vents, but there is no citation to support it. However, later on, some works are mentioned. I suggest adding a citation to this line.

Response: The authors refer the editor and reviewers to page 4, lines 82-85. We have joined the two sentences in order to incorporate the references in the following line to which reviewer three is referring to. 

Line 99: The specific traits are not specified. Which traits are being referred to? It might be better to reorganize this paragraph, starting with the general aspects and then addressing the specific ones.

Response: The authors thank reviewer three for the suggestion. However, we believe we have already done this. The paragraph beginning on page 5 line 100, starts with an introduction to trait-based studies and goes on to define the specific traits that we frame our study within, and refer the readers to table 1 within that block of text on line 110. 

Materials and Methods:

Table 2 is misaligned, please adjust it. Additionally, the sampling equipment used in each zone could also be added.

Response: The authors recognise that the table is misaligned. This has now been changed so that the entire contents of the table fits in the A4 frame of the document. Individual sampling equipment has now been added to a supplementary table containing metadata and accession numbers for all sequences used as table 2 was already very large. 

After Table 2, the paper is not enumerated . I suggest adding the corresponding enumeration.

Response: The authors realise this was a mistake during submission and have corrected this, we apologise for any inconvenience caused during the review process. All documents are now enumerated correctly, and line numbers referenced throughout the response to reviewers. 

In Table 2, it would be helpful to include the sampling method used instead of mentioning it only in the text, or at least briefly mention it.

Response: Please refer to the response above pertaining to adding this information to table 2. 

The samples were not preserved in cold storage. Does this help prevent degradation of their genetic material?

Response: The authors observed over the past ~15 years, that not storing samples that will be used for DNA work in cold storage can affect the quality of the DNA extracted, particularly with regard to genomic or nuclear DNA. It further seems that the degradation effect varies between different faunal groups and could be a result of how “lively” copepods are at the time when they are preserved (Dirivultidae are alive once on board of the ship and stay alive for ~4 days in the fridge, whilst harpacticoids seem to be more sensitive to the change in pressure and/or temperature). The vast majority of the samples came from the Chubacarc cruise in 2019 where samples were immediately fixed in 96 % ethanol and fixed in cold storage (4°C) and an amplification success rate of 77 % was encountered for the Cox1 gene. Samples in 2009 and 2016 were also immediately fixed in cold storage (4C) but have not been stored cold during the entire time after collection. However, all collections are currently at -20°C to ensure preservation of DNA is optimal. 

I would change the title "Species identification" to "Species identification using classical taxonomy."

Response: The authors agree this can be clarified and have changed it to “Taxonomic species identification” for brevity, on page 13, line 223.

 In the Materials and Methods section, it is not recommended to present the results, but species found are mentioned here. I recommend moving this to the Results section. In this section, mention the taxonomic references used and cite them.

Response: The authors agree that these results should not be in the methods and have been moved to the equivalent results section. Please refer to page 21-22, lines 409-415.

Chelex has a standard protocol, what is it? Rather than a "standard protocol," it is the protocol provided by the manufacturer.

Response: The authors have changed “standard protocol” to “protocols provided by the manufacturer”. Please refer to page 15, line, 256.

Both primer pairs had the same annealing temperature. In what cases was each one used? What was it dependent on?

Response: The authors have now clarified in the text that these are not two primer pairs, but a single primer set that has M13 tails attached. This aids in the sequencing of the forward and reverse strands of DNA and are provided with the primers when ordering from Macrogen. This has been clarified in the text. It therefore follows, that there is only one annealing temperature used. Please refer to page 15, 256-261.

“The Cytochrome c. oxidase subunit I gene (Cox1) was amplified using the Folmer [59] primer set LCO1490 (5’GGTCAACAAATCATAAAGATATTGG’3), and CO2198 (5’TAAACTTCAGGGTGACCAAAAAATCA’3). These primers were ordered with corresponding M13 tails attached: M13F-pUC (-40) (5’GTTTTCCCAGTCACGAC’3) and M13R-pUC (-40) (5’CAGGAAACAGCTATGAC’3), for LCO1490 and CO2198, respectively.”

Data Analysis:

It would be convenient to create a table that includes the accession codes of all the sequences used previously and the new ones, along with their taxonomic assignment and the reference of the work in which they were published.

Response: The authors agree. This was also requested by other reviewers and has now been added as an extra supplementary material S1 in the form of an excel spreadsheet (the rest have been relabeled S2-S4 and changed throughout the text)

What type of BLAST was used? Nucleotide or amino acid?

Response: The type of BLAST used was nucleotide. This has now been made clear in the text, please refer to page 16, line 280. 

The delimitation within a species and to determine if a species is cryptic is between 2% and 5%. You mention 20%, which could correspond to differences at the level of different families. This point is very important.

Response: This has now been removed from the text because the 20 % referred to the final alignment. Please see comments above pertaining to this along with the screenshot of what is being referred to by 20 %. The final Dirivultidae alignment had % identity similarity of 80 %. When blasting those sequences initially against nucleotide collections in NCBI, those that did not belong to copepods were removed as these were contaminants. The sequences retrieved from specimens taxonomically identified as S.lauensis that did match with copepods when blasting, all came back as S.lauensis as the first hit with E-values of 0, indicating that, despite a sequence divergence of 20 % in the final alignment, these were unequivocally divergent sequences of S.lauensis and not from other families. Had the latter been true, the species complex would have shown up as a strongly diverging outgroup in the phylogenetic tree. 

Construction of the phylogenetic tree:

Please review references 61 and 65, which do not discuss species delimitation. In general, for delimitation, a tree is input into PTP and the alignment into BEAST to generate an ultrametric tree. In the former, patterns of substitution at the nucleotide level are observed, not at the amino acid level. The latter, BEAST, offers around 4 different models that are the most common. Typically, consensus is used to build trees, without considering the reading frame. However, efforts are made to equalize the length to avoid such problems. I feel that using different models and trees generates more problems when it comes to interpretation.

Response: This was a major concern for reviewer three, however, there is only a misunderstanding in the wording that we clarify here below. 

The authors agree that references 61-65 (page 17, line 300) do not discuss species delimitation. They discuss codon-usage bias and the need for partitioning genes into coding frames prior to phylogenetic tree construction. The references given on page directly after the text pertaining to codon-usage bias as justification for why we partitioned both global alignments. These references are therefore appropriate for the text. The paragraph discussing species delimitation comes later in the text (page 18), and in this paragraph, the appropriate references 67 and 68 are given (page 18, line 331 and 334), which refer to specific delimitation methods. 

In the Bayesian and Maximum Likelihood trees, not all input parameters are indicated, such as the number of iterations, the number of chains, the number of burn-in steps, and the sampling frequency. I believe it is of utmost importance to specify this in the methods.

Response: We thank reviewer 3 for these important and constructive comments. The authors agree that this information was missing from the text and was also requested by another reviewer. This information has now been added to the text to make the analysis reproducible. Please see page 17, 306-316.

“Duplicate sequences were removed prior to tree construction to avoid inflation of support values. Trees were run with 1000 bootstraps and only the highest likelihood tree is displayed. Bayesian posterior probabilities were calculated using unlinked site models, unlinked clock rates, and linked trees (Fig 1 in S3), allowing different evolutionary rates for each partition. Site models were found separately using the Beast Model Test and applied automatically to each partition alongside a strict clock of 1.0. Parameters of the multispecies coalescent were 1.0 for the population mean and linear with constant root for the population function (ploidy was set to mitochondrial). For the priors, the tree construction model was the Yule model, and all standard parameters were retained. A chain length of 200 million, sampling frequency 5000, and burnin of 10 % was used.”

The external groups used should be indicated by their accession code, and I don't find them explicitly mentioned. It would be convenient to explicitly state them in the text.

Response: The authors have now added accession numbers to the text in the paragraph on phylogenetic trees in the methods: page 17-18, lines 318-320. 

“Harpacticoid species were used as the outgroup for the Dirivultidae phylogeny (accession numbers = KX714909, KX714910 and KX714912) and dirivultid species were used as the outgroup for the harpacticoid phylogeny (accession numbers = GQ926008-10), proving to be sufficiently divergent to resolve the trees in both datasets, whilst being sufficiently similar so as to retain topological resolution.”

In the species delimitation analyses, only the methods used are mentioned, but it is not indicated which trees were used or the input values. I believe, once again, that it is important to include this in the Materials and Methods section.

Response: The authors agree that this information is required and has now been added to the text. Please refer to page 18, lines 327-340.

“Species were delimited using a combination of distance-based and tree-based methods and applied to both the global harpacticoid alignment and the alignment containing sequences from S. lauensis from all collections (all other Stygiopontius species have been described and delimited prior to this study). Firstly, the Assemble Species by Automatic Partitioning (ASAP) method [67] was used, which finds the first significant barcoding gap that exists when the divergence between conspecific individuals is smaller than that of individuals of different species from an alignment. Secondly, the Bayesian Poisson Tree Process (bPTP) [68] was used, a process that models branching events in a phylogenetic tree in terms of number of substitutions. This was done using both the ML trees and the BPP trees produced from the alignments in Newick format, and the default parameters from the PTP webserver (MCMC chain = 500,000, thinning =100, and burnin = 10 %). Delimitation results did not differ when using ML or BPP starting trees. Thirdly, Bayes Factor Delimitation (BFD), using nested sampling to calculate the difference between two models, were also used to delimit species from their global alignments.”

Molecular and demographic variance analysis:

The input parameters and the alignment used are not indicated again, given that different models of evolution were calculated.

Response: The authors state in the text on page 19, lines 351-352 that the putative species alignments created from the species delimitation methods were used for the diversity indices calculations and for demographic history reconstruction. This is now made clearer by iterating this on line 360. 

Tree calibration: Where was it extracted from? What bibliographic source supports this? What program was used? What were the other input parameters? Which alignment was used?

Response: We do not understand the reviewers concern as we assume that reviewer three here is referring to (new) page 21, lines 394-397, where we state that the following: 

“A strict molecular clock was calculated based on [19], who report a mutation rate of 1x10-8 events/bp/generation and a generation turnover of 33 generations/year for organisms living at ~20°C. Based on this information, a strict clock of 0.3 mutations/bp/Ma was used to calibrate the coalescent EBSP for each species.”

Here, [19] refers to the following reference: 

19. Gollner S, Stuckas H, Kihara C. T, Laurent S, Kodami S, Arbizu PM. Mitochondrial DNA Analyses Indicate High Diversity, Expansive Population Growth and High Genetic Connectivity of Vent Copepods (Dirivultidae) across Different Oceans. PLoS ONE. 2016 Oct 12;11(10):e0163776. https://doi.org/10.1371/journal.pone.0163776

To answer the reviewers questions in order: 

- Where was it extracted from? 

It was calculated based on the calculations stated in the text from the mutation rate and the generation turnover per year. 

- What bibliographic source supports this? 

The bibliographic source that supports this and from which the clock was calculated is mentioned explicitly in the text. This is reference [19] given above. There are no other references to support this because this is the first time that the generation turnover is mentioned for this group of organisms. The mutation rate used here and taken from [19] is the standard mutation rate for animals and mitochondria. For this reason, the authors are confident that the citation used is appropriate. 

- What program was used?

The authors agree that this was not stated in the text and has now been added. The program used was BEAST2, and the tutorial used to build the EBSP is also stated in this section. Please refer to page 20, line 384-387.

- What were the other input parameters?

The information requested is now explicitly listed on page 21, lines 397-404.

- Which alignment was used? 

This information is stated on page 20, line 386, where the authors state that the EBSP analysis was conducted on each cryptic species, however we have made this clearer by adding the word “alignment” to “each cryptic species…”. 

In citation 28, the number does not allow identification of whom it refers to.

Response: we do not understand the reviewers comment, as the bibliography of ref. 28 is given in the manuscript. 

Results:

Dirivultidae:

It cannot be solely claimed through phylogeny that a species complex exists. If you wish to assert that, I would first cite the results, such as bTPP.

Response: We agree and have added the following line on page 23, line 435: 

“…supported by species delimitation results” 

These species delimitation results are detailed a few lines later page 23, from 440. We are not solely claiming through phylogeny that a species complex exists and therefore have added this to the text to make that clear. 

Is the indicated FST value significant? Was a probability calculated for the FST?

Response: We report all Dxy and Fst values along with significance markings (* for p values <0.05) for Fst values in table 3 but agree that the single p-value for the global Fst calculated from the AMOVA for the S. lauensis group was missing in the text and has been added The Fst value was 0.77 and the p value was 0, and we therefore report it as <0.001 for continuity throughout the document wherever p-values are highly significant. Please refer to line 23, line 450. We add a screen shot of the AMOVA results for further clarification for reviewer three. 

The difference between the Bayesian and Maximum Likelihood trees is due to them being different tree inference methods.

Response: Although the authors fundamentally agree with this, oftentimes the topology of the trees produced by each different method is identical. In this case, either tree is displayed with both ML and PP values shown as ML/PP on the tree, such as in Fig. 3. However, the authors have made changes to the figures so that the trees and haplotype networks appear in the same figure, now colour coded so that the species in the tree match the networks, and the networks are further coloured on a gradient of north to south. The Bayesian posterior probability tree is therefore now in the appendix (S3) along with its corresponding statistics from the BEAST2 run. 

In Figure 2, I cannot distinguish which is the Bayesian probability and which is the maximum likelihood probability.

 Response: We agree that this was difficult to distinguish. This information has been made clearer and is given in the modified caption and legend for this figure, which has now been moved to supplementary information S3. 

Ameiridae:

In Figure 3, when "BEAST three" is mentioned, does it refer to a Bayesian tree? Please specify. 

Response: The authors have restructured the results section (according to reviewer 2 suggestion) and therefore all figures have been renamed. Further, the authors have condensed figures containing phylogenetic trees and networks into two main figures, one for the Dirivultidae and one for the Harpacticoida. Fig. 3 now shows the phylogenetic tree and minimum spanning networks pertaining to the Harpacticoida. The original Fig. 3 referred to just the phylogenetic tree belonging to the Harpacticoida. In the new caption for Fig 3 in the text (page 31, line 623) we write the following:

“Fig 3: Phylogenetic reconstruction of the harpacticoid families associated with vents and corresponding haplotype networks. 

Maximum likelihood tree based on 544 bp of codon position partitioning of the mitochondrial Cox1 for harpacticoid copepods. Trees built with IQ-TREE and BEAST (topologies of ML and BEAST trees were identical and so presented as a single tree). Node values are ML bootstrap/BPP. Species are colour-coded with their corresponding haplotype networks to the right of the phylogenetic tree: B) Ameira sp. 4, C) Ameira sp. 5, D) Sarsamphiascus sp. 1, E) Bathylaophonte pacifica, F) Amphiascus aff. varians 1, and G) Amphiascus aff. varians 2. Networks are coloured on a gradient from North to South. The black symbols under ASAP and bPTP denote the species delimitation by both software (where matching symbols are found, congruence in the species delimitation is inferred by both software, and each symbol corresponds to one species. The grey and black bar denotes whether the species came from shallow water non-vent environments or deep-water hydrothermal vents. There is no mention of “BEAST three” in this text, and as such the authors assume this is a typo from the reviewer. All BEAST trees are Bayesian trees as BEAST stands for Bayesian Evolutionary Analysis Sampling Trees.”

Miraciidae:

The FST value alone does not say much. It is important to indicate the significance, which is not mentioned here. 

Response: The authors agree, and this has now been added to the text (also found in S3, Table 1). Please see page 33, line 676.

Molecular diversity:

Some probability values are not indicated in the FST and AMOVA in the text. Table 3 is not fully displayed.

Response: The authors have revised the manuscript and added p-values to every instance that a Fst value is reported within the molecular diversity sections of each family. Table 3 has been adjusted so that it can be fully displayed. 

Table 4 is not fully displayed.

Response: Table 4 has been adjusted so that it can be fully displayed. 

Discussion:

When it is mentioned that "The Da values are approximately twice as high between S. lauensis and S. aff. lauensis 1 as between S. aff. lauensis 2 and 3, suggesting that divergence took twice as long for the latter two allopatric forms under a constant molecular clock [81]," it is challenging to reach that conclusion if different molecular clocks were not tested or if literature supporting the appropriate molecular clock usage within the genus is not cited.

Response: Please see the response to the comment above pertaining to the molecular clock used and the citation supporting that. The phylogenetic tree was built with a strict clock of 1. A standard procedure used for phylogenetic reconstruction where phylogenetic relationships simply want to be viewed and no time calibration is being conducted. We used a clock of 0.3 to calibrate the EBSP analysis which measures changes in Ne over time. 

14C is not indicative of the food source but rather of time, although 13C could provide information about the food source.

Response: We thank the reviewer for catching our typographical mistake (14C indeed is 13C). 

“These studies elucidate two important findings, firstly that the gut contents of Dirivultid copepods are mostly composed of chemoautotrophic bacteria based on lipid composition and depleted δ15N signatures [52], and secondly, that there is a gradient of specialisation across species in the family, where some members are found very close to the vent source (e.g., S. quadrispinosus) and have evolved to contain high levels of hemoglobin to cope with depleted oxygen levels [23, 104].”

This was a typo and should have said δ13C, referencing [52] which reports δ13C signatures of vent copepods. This has now been changed in the text.

On page 48, lines 1015-1019, we also state: 

“However in in the Okinawa Trough, they were found at both ends of the environmental gradient [51], suggestive of a tolerance to a broader range of environmental conditions than previously thought. The δ15N and δ14C ratios measured from harpacticoids resembled surface sediments but also varied highly, indicating little discrimination in nutritional sources for Ameiridae, Miraciidae and Bathylaophontidae [51].”

This should have said δ13C instead of δ14C and has now been corrected in the text. 

Reference 51 is Nomaki et al. 2019: they investigated dietary sources of meio- and macrofauna at hydrothermal vent fields in the western North Pacific using stable carbon and nitrogen isotope ratios (δ13C, δ15N) and natural-abundance radiocarbon (Δ14C). Reference 52 is Nomaki et al. 2023 who report only δ13C from dirivultid copepods. 

It would be important to mention the issue of working with the selected marker for studying structuring and note that there are other analyses with better resolution, such as the use of microsatellites or mitochondrial genome sequencing.

Response: The authors state the following in the conclusions (page 52, line 1091):

“Future research should include application of genome-wide markers to improve species delimitation and accuracy in determining fine scale structure and demographic patterns in vent copepods.”

To make it clear that the authors agree that the Cox1 gene is not sufficient to draw conclusions about true structure and demography we have added the following:

“However, mitochondrial DNA exhibits a fraction of the true structure and demography of populations and future research should include application of genome-wide markers to improve species delimitation and accuracy in determining fine scale structure and demographic patterns in vent copepods.”

I did not see any comparison of the obtained sequences with sequences from shallower species in the analysis.

Response: As stated in the response to the first comment from reviewer three, this is not the point of the paper. We are not comparing deep-sea hydrothermal vent copepods to shallow water copepods in any quantitative or statistical manner. However, we do compare the sequences phylogenetically and present them in Fig. 3A, in which, deep-sea hydrothermal vent and shallow non-vent clades are differentiated by the grey and black horizontal bar and its legend.

---

## [Decision Letter · Decision Letter 1]

25 Sep 2023

Highly structured populations of deep-sea copepods associated with hydrothermal vents across the Southwest Pacific, despite contrasting life history traits.

PONE-D-23-15095R1

Dear Dr. Diaz-Recio Lorenzo,

We’re pleased to inform you that your manuscript has been judged scientifically suitable for publication and will be formally accepted for publication once it meets all outstanding technical requirements.

Kind regards,

Hans G. Dam, Ph. D.

Academic Editor

PLOS ONE

Additional Editor Comments (optional):

Reviewer 3, who had serious concerns of the first version of the manuscript also reached out via email to let me know that they also found the revised version had addresses all concerns and suggestions, and that the paper was much improved in its presentation. 

Reviewers' comments:

Reviewer's Responses to Questions

**Comments to the Author**

1. If the authors have adequately addressed your comments raised in a previous round of review and you feel that this manuscript is now acceptable for publication, you may indicate that here to bypass the “Comments to the Author” section, enter your conflict of interest statement in the “Confidential to Editor” section, and submit your "Accept" recommendation.

Reviewer #1: All comments have been addressed

2. Is the manuscript technically sound, and do the data support the conclusions?

Reviewer #1: Yes

3. Has the statistical analysis been performed appropriately and rigorously? 

Reviewer #1: Yes

4. Have the authors made all data underlying the findings in their manuscript fully available?

Reviewer #1: Yes

5. Is the manuscript presented in an intelligible fashion and written in standard English?

Reviewer #1: Yes

6. Review Comments to the Author

Reviewer #1: The authors addressed all of my comments in a detailed revision and response.

The editor also asked the reviewers for their views on how the authors addressed all 3 reviewer's questions and concerns. In my opinion, the authors made suitable responses and revisions in regards to the other two reviewer's questions and comments.

7. PLOS authors have the option to publish the peer review history of their article (what does this mean?). If published, this will include your full peer review and any attached files.

Reviewer #1: No

---

## [Editor Report · Acceptance letter]

27 Oct 2023

PONE-D-23-15095R1 

Highly structured populations of deep-sea copepods associated with hydrothermal vents across the Southwest Pacific, despite contrasting life history traits. 

Dear Dr. Diaz-Recio Lorenzo:

I'm pleased to inform you that your manuscript has been deemed suitable for publication in PLOS ONE. Congratulations! Your manuscript is now with our production department. 

Kind regards, 

on behalf of

Dr. Hans G. Dam 

Academic Editor

PLOS ONE